# CD45 pre-exclusion from the tips of T cell microvilli prior to antigen recognition

Yunmin Jung [1✉], Lai Wen [1], Amnon Altman[2,4] & Klaus Ley [1,3,4]

The tyrosine phosphatase CD45 is a major gatekeeper for restraining T cell activation. Its exclusion from the immunological synapse (IS) is crucial for T cell receptor (TCR) signal transduction. Here, we use expansion super-resolution microscopy to reveal that CD45 is mostly pre-excluded from the tips of microvilli (MV) on primary T cells prior to antigen encounter. This pre-exclusion is diminished by depleting cholesterol or by engineering the transmembrane domain of CD45 to increase its membrane integration length, but is independent of the CD45 extracellular domain. We further show that brief MV-mediated contacts can induce $Ca^{2+}$ influx in mouse antigen-specific T cells engaged by antigen-pulsed antigen presenting cells (APC). We propose that the scarcity of CD45 phosphatase activity at the tips of MV enables or facilitates TCR triggering from brief T cell-APC contacts before formation of a stable IS, and that these MV-mediated contacts represent the earliest step in the initiation of a T cell adaptive immune response.

[1] Center for Autoimmunity and Inflammation, La Jolla Institute for Immunology, La Jolla, CA, USA. [2] Center for Cancer Immunotherapy, La Jolla Institute for Immunology, La Jolla, CA, USA. [3] Department of Bioengineering, University of California San Diego, La Jolla, CA, USA. [4] These authors jointly supervised this work: Amnon Altman, Klaus Ley. ✉email: yjung@lji.org

Transmembrane tyrosine phosphatase CD45 is expressed on all nucleated hematopoietic cells and is one of the most abundantly expressed membrane proteins on lymphocytes[1,2]. Analysis of CD45 deficient human or mice showed the important function of CD45 in immune cell development and T cell signaling (reviewed in ref. [3]). While its cytoplasmic domain, including the catalytic domain, shares 95% homology among mammalian species, the highly glycosylated extracellular domain, which includes several splicing variants, shares only 35% homology[4]. The role of each domain has been tested using truncated forms of CD45. For example, the enzymatic activity of cytoplasmic portion of CD45 was sufficient to induce T cell receptor (TCR) signaling[5], while other studies using truncated form of CD45 lacking the extracellular domain showed inhibition of TCR triggering[6,7]. CD45 is a key player in controlling activation threshold of T cell by controlling both positively and negatively regulating lymphocyte-specific protein tyrosine kinase (Lck) activity[8,9]. Recent studies showed that CD45 is critical for restraining the adaptive T cell response by negatively regulating TCR signaling, unless the T cell encounters a high-affinity cognate antigen[10,11].

The molecular mechanism of TCR triggering in the immunological synapse (IS) has been explained by size-dependent kinetic segregation[12,13]. Once TCRs binds to its cognate antigens presented by major histocompatibility complex (MHC) molecules on an antigen presenting cell (APC), CD45 molecules are excluded from the tight contact region, shifting the equilibrium to the phosphorylated state of the immunoreceptor tyrosine-based activation motifs (ITAMs) in the TCR–CD3 complex and allowing productive T cell activation. CD45 has a long extended extracellular domain, which exceeds the length of the TCR–pMHC complex[2]. Therefore, the long extracellular domain of CD45 that cannot be accommodated within the tight contact interface between a T cell and an APC is segregated from the interface[14]. TCR engagement by its cognate MHC-bound peptide is sufficient to drive CD45 exclusion in the absence of any downstream signaling[15]. Recent studies using super-resolution optical techniques have revealed that CD45 exclusion is initiated from microvilli-mediated close contacts that expand as the IS enlarges and matures[16,17]. CD45 exclusion is crucial for initial TCR triggering events[16–18] and it correlates with effective TCR signaling[11,18]. Treatment with a phosphatase inhibitor, pervanadate, or artificially induced CD45 exclusion can initiate TCR triggering in a nonimmune cell reconstitution system[15]. CD45 exclusion also plays a key function in B cell receptor (BCR) signaling and high-affinity immunoglobulin E receptor (FcεRI) signaling in mast cells and basophils[19–21].

Microvilli (MV) are thin actin-based membranous projections that are found on all immune cells[22]. Several recent studies have highlighted the function of MV in T cell activation. TCRs are highly accumulated in the MV of both resting and effector primary T cells, suggesting that MV act as effective sensors for antigenic moieties[23]. Lattice light-sheet microscopy revealed that dynamic motions of MV enable mouse T cells to scan the majority of the opposing surfaces of an APC within 1 min[24]. Quantitative Ca²⁺ imaging combined with interference reflection microscopy (IRM) showed that an intracellular Ca²⁺ response was triggered as early as 1 min after initial contact between a T cell and an activating antibody-coated glass surface, preceding actin remodeling and spreading[25]. Initial MV-mediated isolated contacts and local CD45 exclusion from those areas were sufficient for robust TCR signaling[26]. Thus, MV-mediated initial contacts likely represent an early crucial step in the decision-making process of T cells whether to respond to an antigen.

The plasma membrane (PM) contains diverse types of phospholipids and other lipids such as sterols and sphingolipids, which account for variation in membrane thickness and composition[27–29]. Cholesterol is known to accumulate in the inner membrane leaflet at negative curvatures[30–32], and this high cholesterol content causes an increase in the thickness of the lipid bilayer[27,33–35]. The thickness of pure phospholipid bilayers is ~28–35 Å depending on the types of phospholipid[36,37]. However, the cell membrane at lipid raft domains, which are enriched in sphingomyelin and cholesterol, is 4–8 Å thicker than that of phospholipid bilayers lacking these lipids[27,33,35]. Lipid partitioning in the PM can determine, in turn, the sorting of membrane proteins[38]. It has been reported that ~95% of CD45 is found in the non-raft fraction of cell membrane[39].

In this work, we combine the 4x expansion microscopy[40,41] and Airyscan laser scanning confocal imaging[42], which yields an effective resolution of 30–40 nm. We report that CD45 is mostly pre-excluded from the tips of MV in T cells prior to their early searching and scanning of APCs, potentially as a result of diffusion barriers formed by cholesterol accumulation and, hence, thicker membrane at the tip region. We confirm this pre-exclusion of CD45 at the tips of MV using an independent approach, stochastic optical reconstruction microscopy (STORM). This CD45 pre-exclusion is found in different types of lymphocytes, including CD4⁺ and CD8⁺ T cells, regulatory T (Treg) cells, and B cells. Furthermore, MV-mediated engagement of APCs by antigen-specific T cells during the early scanning stage results in rapid Ca²⁺ influx.

## Results

**CD45 is pre-excluded from the tips of MV in human and mouse T cells.** Under conventional microscopy, the thin and short protrusive structures of MV appear as blurred short spikes on the cell surface. Therefore, mapping the distribution of molecules of interest in relation to MV and their subregions requires super-resolution imaging techniques. Although Airyscan imaging offers ~1.7-fold better resolution than that achieved by conventional light microscopy[42], Airyscan was not sufficient to resolve the localization of labeled CD45 on the MV (Fig. 1a). To achieve better resolution, we applied 4x expansion microscopy[40,41] combined with Airyscan confocal imaging (4x-ExM-Airyscan). Since the resolving power of expansion microscopy linearly increases as a function of the expansion factor of the swellable hydrogel, a combination of these two techniques can achieve a 6.8 ($1.7 \times 4$)-fold higher resolution than the diffraction limit. This approach allowed imaging of the entire cell, making it possible to obtain cross-sectional images of hundreds of MV, thus enabling us to unravel the nanoscale molecular distribution of CD45. We found that CD45 was largely excluded from the tips of MV on freshly isolated, unstimulated human CD4⁺ T cells (Fig. 1b, yellow arrows). Post-staining with the membrane staining dye, FM 4-64FX, confirmed that CD45 was largely excluded from the tips of MV (Fig. 1c). To rule out the possibility that this exclusion was caused by incomplete expansion of the gel at the tip region, we imaged the localization of L-selectin (L-sel), which is known to be expressed on T cell MV[23,43,44], and found that, unlike CD45, which was largely excluded from the tips, L-sel was highly concentrated at the tips, as reported previously[23,43,44] (Fig. 1d–f, Supplementary Fig. 1 and Supplementary Fig. 2a–c). CD45 exclusion did not correlate with the expression level of L-sel (Supplementary Fig. 3a–d). To quantify the molecular distributions, we firstly calculated the relative intensities of L-sel and CD45 molecules by determining the ratiometric intensity score (RIS) (Supplementary Table 1, Eq. (1)), originally introduced as generalized polarization (GP) for comparing the relative intensities of two channels, X and Y[45]. We normalized Y intensity to match the mean intensities between X and Y and named it RIS. RIS becomes zero when both molecules have the

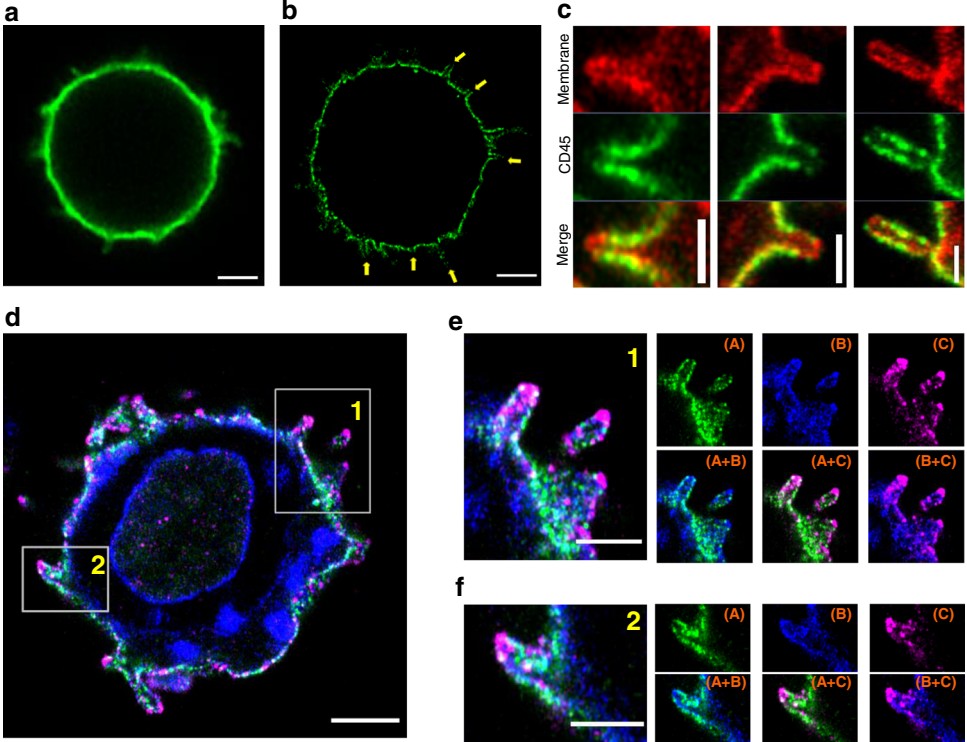

**Fig. 1 Pre-exclusion of CD45 at the tips of MV on human CD4+ T cells. a** A representative Airyscan image of a human CD4+ T cell labeled with Alexa Fluor 488 (AF488) conjugated anti-CD45 antibody. **b** A representative 4x-ExM-Airyscan image of a similarly labeled cell. Yellow arrows indicate MV tips. **c** Representative 4x-ExM-Airyscan images of MV on human CD4+ T cells stained with anti-CD45-AF488 (green) and FM 4-64FX membrane dye (red). **d** A representative 4x-ExM-Airyscan image of a human CD4+ T cell stained with anti-CD45-AF488 (green), FM 4-64FX membrane dye (blue) and anti-L-sel-AF568 (magenta). **e, f** Magnified images of areas 1 and 2 marked by the rectangles in **d**. Individual channel images of CD45 (A), membrane (B), and L-sel (C) (right, upper panels) and merged images of (A) and (B), (A) and (C), and (B) and (C) (right, lower panels). These images are representative of nine cells observed in three independent experiments. Scale bars in **a**, **b**, **d**: 2 μm; in **c**: 500 nm; **e**, **f**: 1 μm. The scale bars are corrected by the expansion factor except for **a**.

same normalized intensities, +1 when X is only present, and −1 when Y is only present. Here we set CD45 (X) and L-sel (Y) so that RIS value of −1 represents 100% exclusion of CD45, and +1 represents 100% exclusion of L-sel. The color coded RIS image where yellow is −1 and blue is +1 showed the L-sel is indeed significantly concentrated at MV tips whereas CD45 was highly excluded from the tips (Supplementary Fig. 2d). Next, we segmented subareas of the cell to assess the molecular distributions of L-sel vs. CD45 at each segmented area. Supplementary Fig. 2e shows each tip points (MV tips, red cross) and the segmented areas for tip region, in which pixel distances were <150 nm from the tip (MV-tip area, yellow), MV-column area (MV-col area, cyan), and cell body area (CB area, orange). The L-sel and CD45 intensity scatter plots showed that in MV-tip area, L-sel was mostly positive while CD45 was very low. The CD45 intensities increased in MV-col area and CB area while the L-sel intensities decreased in the same areas (Supplementary Fig. 2f). The median intensities of CD45 and L-sel in each segmented area of L-sel^high resting T cell were analyzed in Supplementary Fig. 2m, n, respectively. The mean of the median intensities of CD45 in the MV-col area was 5.4-fold higher than in the MV-tip area (Supplementary Fig. 2m). The RIS values were highly negative in the MV-tip area and it were increased in the MV-col area and the CB area (Supplementary Fig. 2o).

To determine whether CD45 exclusion was robustly detected independently of intensity variations, we applied a method that we termed segment-correlation coefficient, SR′, between the signals from two channels (Supplementary Table 1, Eq. (3), Supplementary Note, and Supplementary Fig. 4). SR′ is a modified version of Pearson's correlation. Pearson's correlation

(R′) (Supplementary Table 1, Eq. (2)) is insensitive to the correlation of the segmented subarea, for example, if all pixel intensities of X in the segmented area are 0 (100% exclusion of molecule X) while all pixel intensities of Y in the segmented area are equal to maximum intensity, (100% enriched molecule Y), the variance of $(X_i − \bar{X})(Y_i − \bar{Y})$ (Supplementary Table 1, Eq. (2)) become zero, thus R′ is zero instead of −1. SR′ quantifies the contribution of negative or positive image correlation between the two-channel intensities within a segmented area, regardless of variations in signal intensities among cells. SR′ values from simulated data for validation (Supplementary Note and Supplementary Fig. 4a–f) and from the segmented data by distance or RIS values of a representative cell (Supplementary Fig. 4j, k) showed that SR′ faithfully displayed a correlation between two channels within the segmented area, and it was sensitive to the RIS between the channels. The SR′ of the MV-tip area was mostly negative compared to the MV-col area or the CB area (Supplementary Fig. 2p). Note that the SR′ of the entire area (Cell) is equal to the Pearson's correlation, R′. Since the RIS values near the tips were mostly negative (Supplementary Fig. 2d, o), this anticorrelation of SR′ indicates that CD45 molecules are highly depleted at the MV tips while L-sel molecules are accumulated.

Membrane lipid domains are sensitive to temperature, and temperature effects are largely dependent on the composition of lipids, particularly cholesterol[46–49]. For instance, large scale lipid phase separation was disrupted at 37 °C in a model membrane[50], and cold-induced T cell activation was reported[51]. It was also reported that the liquid-ordered state of lipid domains is not

disrupted at physiological temperature[52]. Several studies also showed that lipid domains became extremely stable at ~37–46 °C in the presence of high cholesterol or glycosphingolipids concentrations[48,49,53]. To test whether CD45 exclusion was sensitive to temperature, we labeled cells with antibodies at 37 °C for 15 min. Although the intensities of CD45 (Supplementary Fig. 2m) and L-sel (Supplementary Fig. 2n) were lower in cells labeled at 37 °C in MV-tip area, analysis of SR′ in the MV-tip area showed no significant differences between samples prepared at 4 and 37 °C. Thus, we concluded that CD45 exclusion from the MV tips was not caused by cooling to 4 °C.

We confirmed CD45 pre-exclusion from the MV tips by using another super-resolution imaging technique, 3D-STORM microscopy (Supplementary Fig. 5). 3D-pair-correlation[54] between the L-sel and CD45 was calculated with the data within 9 × 9 pixels around tip positions. The pair-correlation between X and Y molecules displays the probability of finding an X molecule at a given radius ($r$) from the Y molecule. Here we calculated the probabilities of finding CD45 molecules at given radius from L-sel molecules and the result displayed a clear peak at ~50 nm (Supplementary Fig. 5e), which matches the radius of MV[22,23].

We observed similar CD45 pre-exclusion in human CD8+ T cells, (Supplementary Fig. 6a–e) and mouse CD4+ T cells (Supplementary Fig. 6f–j), as well as in Treg cells sorted from Foxp3[YFP-Cre] transgenic mice (Supplementary Fig. 6k–o). These findings indicate that CD45 pre-exclusion from the tips of MV is a universal phenomenon in T cells. Importantly, this CD45 pre-exclusion from the tips of MV in unstimulated T cells was distinct from the previously reported ligand-induced exclusion[17].

**CD45 is mostly excluded from the tips of MV where TCRs are concentrated.** TCRs are accumulated in MV of quiescent human and mouse T cells[23,24]. Combining these findings with our observation of CD45 pre-exclusion, TCRs that are relatively free from adjacent CD45 are expected to be found at the tips of MV. As shown in Fig. 2a–c and Supplementary Movie 1, CD3 molecules were concentrated on the MV compared to the CB, while CD45 molecules were largely excluded from the tips. The RIS values were mostly negative at the MV tips (Fig. 2d). The CD3 and CD45 intensity scatter plots for the corresponding segmented areas (Fig. 2e) showed that CD45 was largely depleted in the MV-tip area while the level of CD3 in this area was high. CD45 intensities were increased in the MV-col area and the CB area (Fig. 2f).

Next, we correlated the spatial distribution of intensities, RIS, and SR′ values to the position of MV tips at a higher resolution. We segmented pixels by distance from tips with a bin size of 12.5 nm, and calculated intensities, RIS and SR′. Since the intensity variation between the cells were large, the intensities of each cell were normalized between 0–1 after saturating the upper 1% of the intensity. The median intensities of CD45 (Fig. 2g) were close to zero at the tips and the median CD3 intensities (Fig. 2i) were mostly high at the tip positions (where 0 is the position of tips). The RIS (Fig. 2k) and SR′ plots (Fig. 2m) were mostly negative at the tip positions. Intensities, RIS and SR′ were also calculated for the segmented areas shown in Fig. 2e. CD45 intensities at the MV-tip area (Tip) were significantly lower than at the MV-col area (Col) and the CB area (CB) (Fig. 2h). The segmented CD3 intensities at the MV-tip area were significantly higher than in other areas (Fig. 2j). The RIS values of the MV-tip area were significantly negative while those of the MV-col area were close to zero and those of the CB area became positive (Fig. 2l). Similarly, the mean SR′ values of the MV-tip area were the most negative and were significantly different between the other areas. Since the RIS values near the tips were mostly negative (Fig. 2d, k, l), this

anticorrelation of SR′ between CD45 and CD3 clearly indicates that TCRs localized at the tips of MV are mostly free from the adjacent CD45 molecules.

**BCR is accumulated at the tips of MVs and CD45 is largely excluded.** Next, we investigated whether CD45 pre-exclusion is found in B cells. Similar to T cells, CD45 pre-exclusion from the MV tips was also found in both human and mouse B cells (Supplementary Fig. 7a, b). We also investigated the distribution of the BCR vs. CD45 in B cells. A previous study showed that ICAM1 and MHC-II molecules are present on MV while LFA1 and CD40 are excluded[55]; however, BCR localization in MV has not been reported. We found that CF633-labeled BCR (IgD or IgM) molecules on the cell surface were highly concentrated on MV (Fig. 3b and Supplementary Fig. 7d). Similar to TCRs, BCR molecules were highly accumulated at the tips of MV, where CD45 molecules were largely excluded (Fig. 3 and Supplementary Fig. 7). These results indicate that TCRs and BCRs at the tips of MV are mostly devoid of CD45 molecules.

**Depletion of cholesterol diminishes the pre-exclusion of CD45 from the MV tips.** It has been reported that the presence of cholesterol is a major factor that affects the membrane thickness[27,33,35]. We hypothesized that the accumulation of cholesterol at the MV tips causes local thickening of the PM at the tips, which in turn, could create a diffusion barrier for CD45. To test this hypothesis, we took two independent approaches, i.e., depleting cholesterol from the cell membrane using methyl-β-cyclodextrin (MβCD), and genetically altering the CD45 trans-membrane domain. MβCD treatment is known to disrupt lipid rafts in human primary T cells and Jurkat T cells[56–58]. When we treated freshly isolated (resting) or effector human CD4+ T cells with 10 mM MβCD for 30 min, we found more than 90% of both resting and effector CD4+ T cells were still viable after treatment (Supplementary Fig. 8a–d). MβCD treatment increased the mean CD45 intensities in effector T cells, but had no significant effect on the mean intensity of CD45 and CD3 in resting T cells or CD3 in effector T cells (Supplementary Fig. 8e). Analysis of the intensity changes within the segmented areas showed that the increments of CD45 in MV-tip areas in both MβCD- treated resting and effector T cells were significant (Supplementary Fig. 9a, c).

We quantified the RIS and the SR′ for the segmented areas (Supplementary Fig. 9e–h, left) as described earlier and compared the results between the untreated and MβCD-treated samples. The RIS of the MV-tip areas in both resting (Fig. 4e) and effector CD4+ T cells (Fig. 4h) were increased upon MβCD treatment. Also, the intensity scatter plots for the segmented areas (Supplementary Fig. 9e–h, right) corresponding to the cells in Fig. 4a–d showed clear alterations of CD45 and CD3, especially in the MV-tip areas, upon MβCD treatment. The intensity ratios of the CD45 at the MV-col area vs. MV-tip area were also decreased upon MβCD treatment from 6.4-fold to 1.9-fold for the resting T cells (Supplementary Fig. 9a) and from 8.6-fold to 1.7-fold for the effector T cells (Supplementary Fig. 9c). These results indicate that CD45 exclusion from MV tips is mitigated by MβCD treatment. Consistent with this, the correlation values between CD45 and CD3 near the tip area calculated by SR′ for the segmented areas were significantly increased upon MβCD treatment in all areas compared to untreated cells. (Fig. 4f, h). These results support the conclusion that cholesterol depletion diminishes the pre-exclusion of CD45 from the MV tips.

**The CD45 short membrane integration limit accounts for its pre-exclusion.** We considered the possibility that the relatively

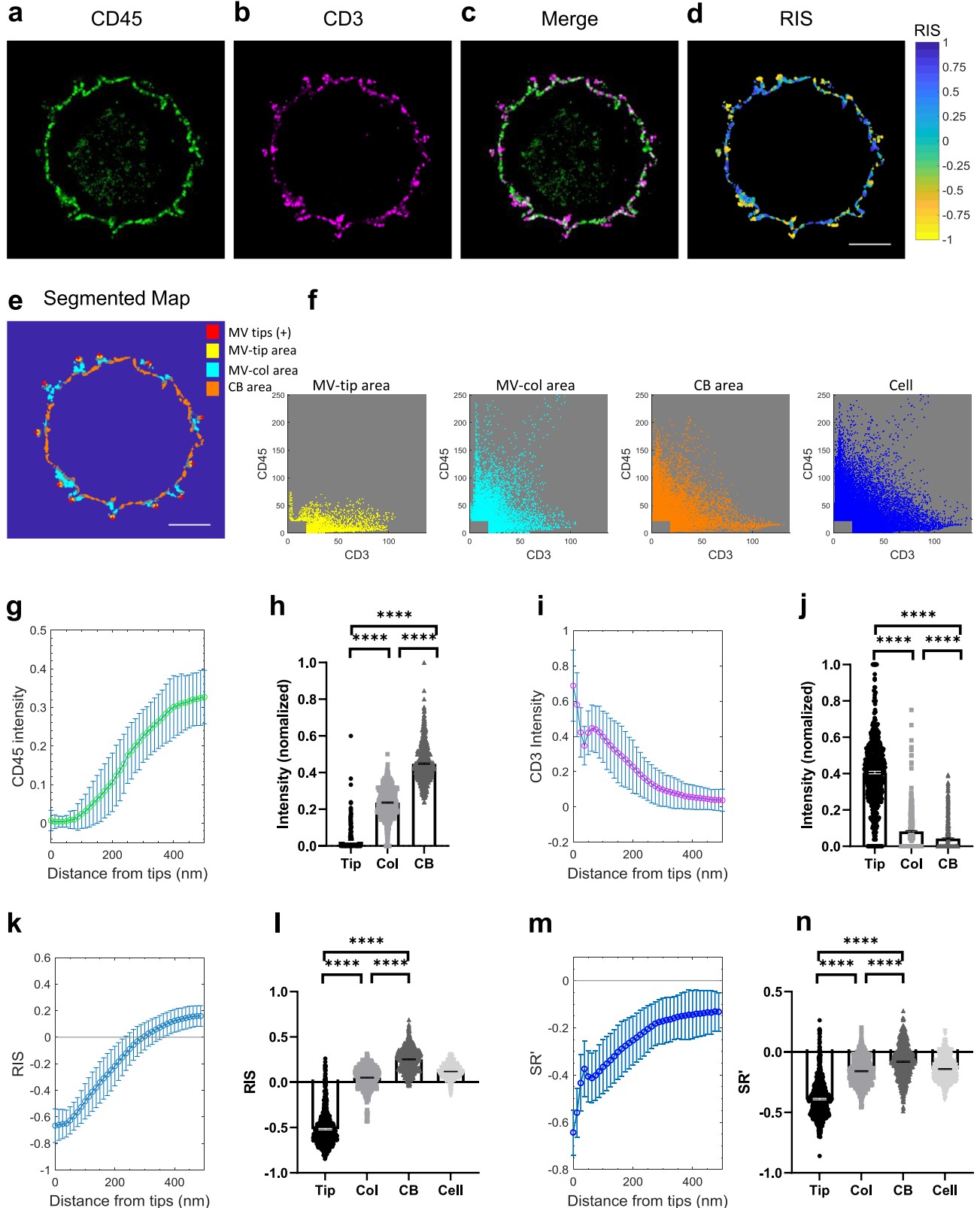

short membrane integration limit (MIL) of CD45 may account for its exclusion from the thicker PM at the MV tips. We define MIL as the maximum number of amino acids (a.a.) between two charged residues [(lysine (K), arginine (R), glutamic acid (E), aspartic acid (D), and histidine (H)] located nearest to each end of the α-helix TM domain of a PM-residing protein

(Supplementary Table 2). For instance, as shown in Fig. 5a, the predicted TM domain lengths of both CD45 and CD3δ are 21 a.a. However, the MIL of CD45 is 22 a.a., flanked by positively charged lysine (K) residues at each end, while that of CD3δ is 27 a.a. between aspartic acid (D) and histidine (H). We here define the extra a.a. residues within the MIL outside the TM domain as

**Fig. 2 Presence of CD3 molecules on tips of MV, where CD45 is largely depleted, in human CD4$^+$ T cells. a–c** Representative 4x-ExM-Airyscan images of a CD4$^+$ T cell labeled with anti-CD45-AF488 (green; **a**), anti-CD3-CF633 (magenta; **b**), and the merged image (**c**). Images in **a–c** are representative of 35 cells observed in three independent experiments. **d** The RIS image between **a** and **b**. **e** The segmented map of the cell in **a–c** for positions of the individual MV tips (MV tips, red cross), MV-tip area (yellow), MV-column area (MV-col area, cyan), and cell body area (CB area, orange). Scale bars in **d** and **e**: 2 μm. **f** The intensity scatter plots for CD3 and CD45 within the MV-tip area (yellow), MV-col area (cyan), CB area (orange), and entire cell area (blue) of the cell shown in **e**. **g** The mean of the median CD45-intensities (normalized) of 35 cells was plotted as a function of the distance from the MV tips. Error bars represent the standard deviation (SD). **h** The median CD45-intensities (normalized) within the segmented areas (MV-tip area (Tip, black circle), MV-col area (Col, gray square), CB area (CB, dark gray triangle)) of each z-plane images (20 z-plane images per cell) of the 35 cells. Each dot represents data collected from a z-plane image. Bars represent the mean and error bars represent the standard error of the mean (SE). **i** The mean of the median CD3 intensities (normalized) was analyzed as **g**. **j** The median CD3 intensities (normalized) within the segmented areas described as **h**. **k** The mean of the median RIS values between the CD45 and the CD3 images was analyzed as **g**. **l** The median RIS values within the segmented areas or entire cell (Cell, light gray downward-triangle) described as **h**. **m** The mean of SR′ values between the CD45 and the CD3 images was analyzed as **g**. **n** The mean SR′ values within the segmented areas described as **l**. *p* values (****$p \leq 0.0001$) were calculated by two-tailed Wilcoxon matched-pairs signed rank test. Source data for **g–n** are provided as a Source Data file.

spacer. Since charged a.a. are most unlikely to be integrated into the hydrophobic core in the membrane lipid bilayer[59,60], this MIL represents the maximal number of a.a. (spacer plus TM domain) that can theoretically be integrated into the membrane.

The charged a.a. residues at position K$^{577}$ and K$^{600}$ of human CD45 are 100% conserved among 38 different species (Fig. 5b and Supplementary Table 3), and the MIL of these CD45 molecules (22 a.a.) is shorter than that of most other major TM proteins expressed on T cells, B cells, or APCs (Fig. 5c, d and Supplementary Table 2). The mean MIL length between charged a.a. (K, R, E, D, H) of the non-CD45 membrane proteins (Supplementary Table 2) is 27.4 a.a, which is predicted to be a 41.1 Å-long α-helix [1.5 Å per residue[61]], while the 22 a.a. MIL of CD45 is predicted to be only a ~33 Å-long α-helix, which matches the thickness of a pure phospholipid bilayer, but is substantially shorter than the thickness of the membrane at lipid raft domains. Therefore, we hypothesized that CD45, which has a shorter MIL, cannot diffuse into the thicker membrane at the MV tips, resulting in its exclusion, while other proteins that have a longer MIL can.

To test whether the shorter MIL of CD45 and/or its long extracellular domain were responsible for its relative pre-exclusion from the MV tips, we constructed two CD45 mutants with an added N-terminal triple hemagglutinin A (HA) tag sequence (Fig. 6a): CD45ΔEC, from which the extracellular domain of human CD45 was deleted except for six a.a. adjacent to the TM domain; and CD45ΔECMIL25, in which the lysine (K) at position +23 in the C-terminus of the TM domain in CD45ΔEC was replaced with a hydrophobic a.a., leucine (L), thereby increasing its MIL from 22 in the wild-type CD45 TM domain to 25 a.a. (Fig. 6a). We then determined the distribution of these mutant CD45 molecules in comparison to that of endogenous CD45 in lentivirally transduced Jurkat T cells by staining with anti-HA or anti-CD45 antibodies, respectively. The cells properly expressed their endogenous CD45, as well as the transduced CD45 mutants (Fig. 6b, c and Supplementary Figs. 10 and 11).

The CD45ΔEC-transduced cells displayed clear depletion of the mutant CD45 at the MV tips similar to that of the endogenous CD45 (Fig. 6b and Supplementary Figs. 10a and 11a), indicating that CD45 pre-exclusion from the tips of MV was not driven by the long extracellular domain. In sharp contrast, the CD45ΔECMIL25-transduced cells no longer displayed exclusion of the mutated CD45 from the MV tips (Fig. 6c and Supplementary Figs. 10b and 11b). The negative RIS values found at the MV tips indicated that the CD45ΔECMIL25 mutant was neither restricted to, nor excluded from, the MV tips. Because the expression levels of both CD45ΔECMIL25 and endogenous CD45 were low and discontinuous on the surface, which made it difficult to define the location of tips, we had to modify the

analysis method from that used in primary cells (i.e., Fig. 2e–n). We selected 120 MV images each collected from the CD45ΔEC- or CD45ΔECMIL25-transduced cells (Supplementary Fig. 11) and analyzed the Pearson's correlation coefficient (R′), Manders' overlap coefficient (MOC), and Manders' correlation coefficients (MCC$_1$ and MCC$_2$) (Supplementary Table 1)[62] between endogenous and mutant CD45. As shown in Fig. 6d, those colocalization coefficients were significantly reduced in CD45ΔECMIL25-transduced cells compared to CD45ΔEC-transduced cells. Thus, we conclude that the increased MIL of the TM domain in the CD45ΔECMIL25 mutant, rather than the extracellular domain, enabled this CD45 mutant to diffuse into the membrane of MV tips and, therefore, that the short MIL of the native, endogenous CD45 TM domain is indeed likely responsible for CD45 pre-exclusion from the MV tips.

**Early MV-mediated interactions between T cells and APCs induce Ca$^{2+}$ influx.** To determine the functional significance of CD45 pre-exclusion, we tested whether MV-mediated transient contacts with APCs can trigger early TCR signaling, measured by Ca$^{2+}$ influx, prior to the formation of a stable IS, at which time CD45 exclusion from tight T cell-APC contacts is driven by size exclusion[16,17]. We predicted that the absence of (or lesser) inhibition of TCR signaling at the tips of MV at this early time due to CD45 pre-exclusion may enable or facilitate local Ca$^{2+}$ triggering from brief T cell-APC contacts before the organization of a mature, flattened IS. T cell MV are known to scan the APC surface[24], but it was not known whether these transient MV-mediated contacts can induce Ca$^{2+}$ influx in T cells. It has been reported that actin-rich protrusions coupled to contact to antigen-coated surface initiated T cell signaling[63,64] and a recent study by Fritzsche et al. using a combination of IRM and fluorescence imaging also demonstrated immediate calcium influx initiated in Jurkat T cells by minute contacts with an activating antibody-coated glass surface[65]. However, to our knowledge, direct 3D-live imaging of MV-mediated contacts between antigen-specific T cells and APCs initiated by TCR signaling has not been reported. We applied fast sub-diffraction limit 3D optical diffraction tomographic (ODT) live cell microscopy to address this question[66]. We used an antigen-specific system, OTII x *Rag2*$^{-/-}$ (KO) TCR-transgenic (OTII-*Rag2*$^{-/-}$) CD4$^+$ T cells specific for ovalbumin (OVA), which were loaded with the Ca$^{2+}$ indicator Fluo-4 AM and stimulated with OVA peptide-pulsed B cell APCs. The maximum theoretical lateral resolution of the ODT system can reach ~110 nm[67,68], although the practical lateral resolution of the system was reported to be 172 nm[69]. As shown in Supplementary Movies 2 and 3, T cells established contacts with OVA-pulsed B cells via their MV, and these brief contacts induced Ca$^{2+}$ flux in T cells. OTII-*Rag2*$^{-/-}$ CD4$^+$

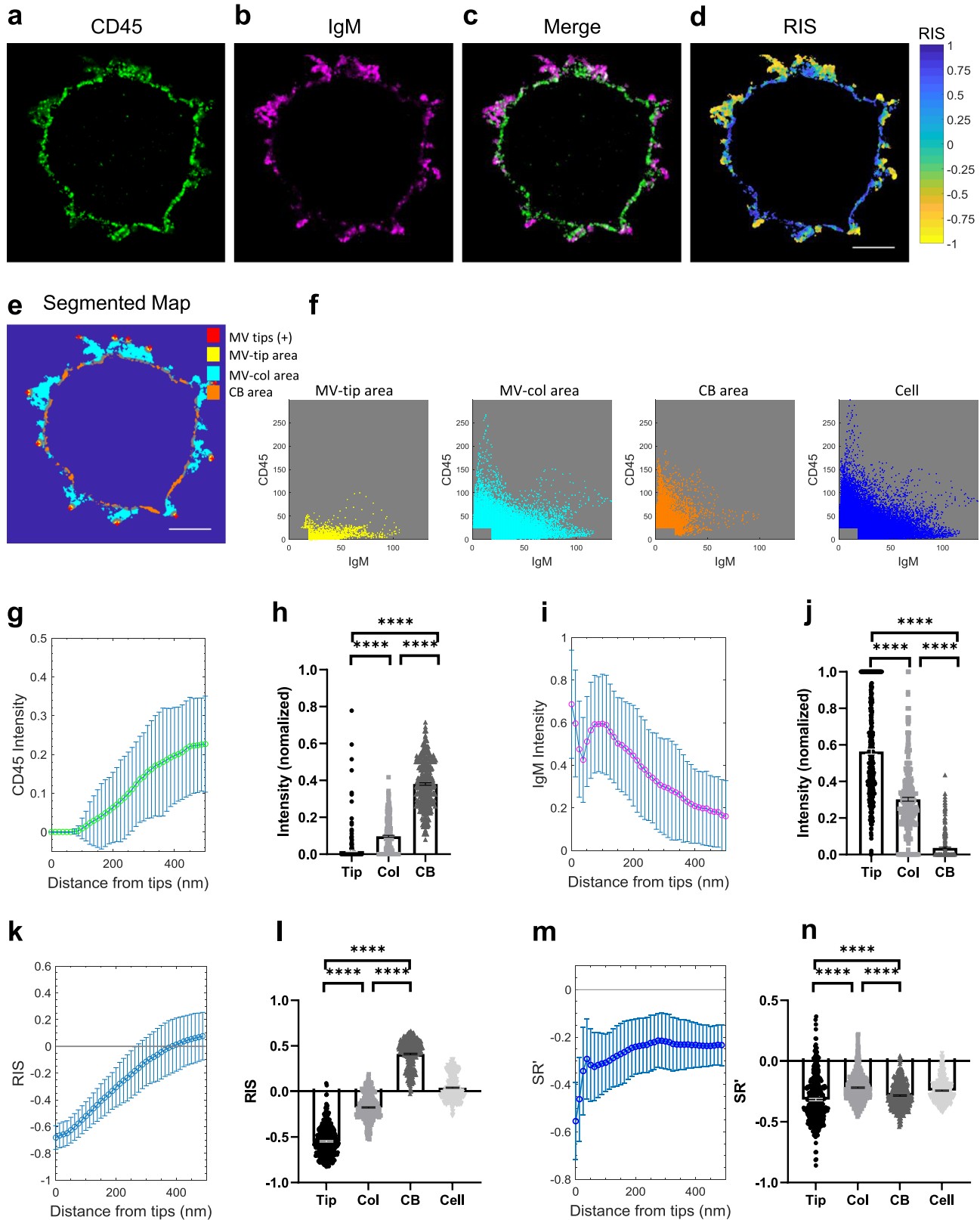

T cells mixed with B cells that were not pulsed with antigen did not induce calcium flux. (Supplementary Movie 4). The mean of the intensities of Fluo-4 AM in T cells that persistently contact an antigen-pulsed B cell (contact) during the acquisition time (5 min) was significantly higher than that of T cells that made no detectable contact with any other cell (no-contact) in the presence of antigen-pulsed B cells (Supplementary Fig. 12a). Note that we did not observe completely flattened T-B cell interfaces in T-B cell conjugates; rather, the persistent contacts are mediated by dynamic movement of MVs on both contacting T and B cells, indicating that brief MV-mediated contacts effectively induce $Ca^{2+}$ signals. We also quantified the intensity changes

**Fig. 3 Highly enriched membrane-bound IgM molecules on the tips of MV of human B cells. a–c** Representative 4x-ExM-Airyscan images of a B cell labeled with anti-CD45-AF488 (green; **a**), anti-IgM-CF633 (magenta; **b**), and the merged image (**c**). Images in **a–c** are representative of 19 cells observed in three independent experiments. **d** The RIS image between **a** and **b**. **e** The segmented map of the cell in **a–c** for positions of the individual MV tips (MV tips, red cross), MV-tip area (yellow), MV-col area (cyan), and cell body area (CB area, orange). Scale bars in **d** and **e**: 2 μm. **f** The intensity scatter plots for IgM and CD45 within the MV-tip area (yellow), MV-col area (cyan), CB area (orange), and entire cell area (Cell, blue) of the cell shown in **e**. **g** The mean of the median CD45-intensities (normalized) of 19 cells was plotted as a function of the distance from the MV tips. Error bars represent the SD. **h** The median CD45-intensities (normalized) within the segmented areas (MV-tip area (Tip, black circle), MV-col area (Col, gray square), CB area (CB, dark gray triangle)) of each z-plane images (20 z-plane images per cell) of the B cells (19 cells). Each dot represents data collected from a z-plane image. Bars represent the mean and error bars represent the SE. **i** The mean of the median IgM-intensities (normalized) was analyzed as **g**. **j** The median IgM-intensities (normalized) within the segmented areas described as **h**. **k** The mean of the median RIS values between the CD45 and the IgM images was analyzed as **g**. **l** The median RIS values within the segmented areas or entire cell (Cell, light gray downward-triangle) described as **h**. **m** The mean SR′ values between the CD45 and the IgM images was analyzed as **g**. **n** The mean SR′ values within the segmented areas described as **l**. *p* values (****$p \leq 0.0001$) were calculated by two-tailed Wilcoxon matched-pairs signed rank test. Source data for **g–n** are provided as a Source Data file.

(I − I(0)) relative to the intensity at time zero, found that $Ca^{2+}$ signals were maintained for at least 5 min during the contacts (Supplementary Fig. 12b, c). We were able to observe cells that initiated contacts during the acquisition time, showing that the Fluo-4 AM intensities of seven out of ten cells increased upon making contacts (Supplementary Fig. 12a, d). The $Ca^{2+}$ signals often fluctuated during the MV-mediated contacts. Thus, the $Ca^{2+}$ flux oscillations previously observed during initial T cell-APC contacts[70] were likely due to the brief early contacts made by MV, which could not be resolved in previous studies.

## Discussion

Here we report a distinctive feature of T cell MV, namely, CD45 pre-exclusion from their tips prior to engagement by APCs. This pre-exclusion was observed in human and mouse lymphocytes, including CD4[+], CD8[+] T cells, Treg cells, and B cells. This spatial pre-exclusion of CD45 resulted in more CD45-free TCRs on the MV tips. The CD45 pre-exclusion from the MV tips was likely caused by the short MIL (22 a.a.) of the CD45 TM domain that prevents CD45 from diffusing into the thicker PM at the tips of MV due to the higher cholesterol content at its negative curvature.

To quantify the depletion of CD45 from MV tips, it was necessary to define the location of tips. We first tested the fixable membrane dye, FM 4-64FX. Although FM 4-64FX did label the membrane and allowed multicolor application, its membrane labeling was often discontinuous and it also showed strong intracellular staining (Fig. 1d). We therefore co-labeled molecules that are known to be found at the MV tips, namely CD3[23] or L-sel[43,44]. The composite images of CD45 with CD3 or L-sel allowed us to observe the complete morphology of MV and made it possible to determine the location of tips by adopting the convex hull algorithm[71].

It has been reported that CD45 exclusion from the IS was initiated from isolated tight contacts originating from MV[16,17]. Here, we show that CD45 is actually pre-excluded from the MV tips before forming a tight contact with APCs. A theoretical model of TCR triggering suggested that two factors are key to initiating TCR signaling: (i) the dwell time of TCRs in the CD45-depleted area; and (ii) spatial constraints on contact area size imposed by cell topography[72]. Another recent study, which used nanofabrication methods to prevent CD45 exclusion mediated by its extracellular domain, showed that TCR signals were induced when small TCR clusters are formed (inter-TCR spacing of <50 nm)[73]. These studies indicate that the spatial distribution of TCR and CD45 both play a key function in controlling the triggering threshold.

Our findings that TCRs are present in a CD45-depleted zone at small constrained areas of the MV tips is consistent with the

model accounting for how TCR sensitivity is maintained while avoiding non-specific activation. CD45 pre-exclusion has distinct consequences: (1) The relative absence of this long molecules at or near the tip of MV facilitates scanning by TCR and engagement of APCs presenting pMHC complexes, or else TCRs will be inaccessible for engagement due to the taller neighboring CD45 molecules; and (2) The absence of inhibitory CD45 phosphatase activity enables effective initiation of TCR signaling.

CD45 pre-exclusion discovered herein suggests an earliest TCR signaling, for example, the ability of CD45-free TCRs at the tips of MV to effectively scan the surface of APCs and transduce signals. This clearly happens prior to the formation of a mature IS and may nucleate CD45 exclusion from the IS, which progresses as the T cell-APC contact area is enlarging. During the IS formation process, the collapse of MV may allow additional TCR molecules to be recruited from the columns of MV to the IS, and CD45 to be further excluded from the flattened interface due to its long extracellular domain[13,14].

Our study demonstrates that $Ca^{2+}$ influx is induced by the MV-mediated contacts prior to the formation of a mature, flattened IS. Recent studies showed that CD45 segregates from isolated tight MV contacts within seconds of TCR engagement, and that increased TCR phosphorylation correlates with this TCR-CD45 segregation[17,26]. Several super-resolution studies reported preexisting clusters of TCR and other major signaling molecules such as LAT, Lck, ZAP-70, or SLP-76 prior to activation[23,74–76]. Moreover, a recent study showed that the TCR co-receptors, CD4 and CD2, early signaling molecules, Lck and LAT, and ERM (ezrin, radixin, and moesin) proteins are highly expressed on MV[77]. These findings suggest that the preexisting clusters could be related to the surface topography. Together, these findings implicate MV as specialized cellular domains where initial antigen recognition and signal transduction occur. Thus, TCRs on the tips of MV are ready to trigger upon engagement of pMHC, whereas TCRs at other MV areas need to be segregated from CD45 during IS maturation.

We hypothesized that the molecular mechanism of CD45 pre-exclusion at the MV tips may be related to the curvature-driven lipid partitioning of the PM at the tips of MV and the limited length of the CD45 TM domain. It has been suggested that the highly curved membrane at the tips of MV can lead to local sorting of specific lipids[31]. Although the specific lipid composition at the protrusive membrane regions is unknown, studies in model membranes have shown that cholesterol accumulation at highly negative curvatures promotes membrane stability[78]. Cholesterol is known to stabilize the membrane lipid domains at physiological temperature[46–49]. We found that CD45 pre-exclusion from the MV tips is stable at physiological temperature (Supplementary Fig 2g–p). This stability would promote the residence of cholesterol-rich domains at the tips of MV. A recent

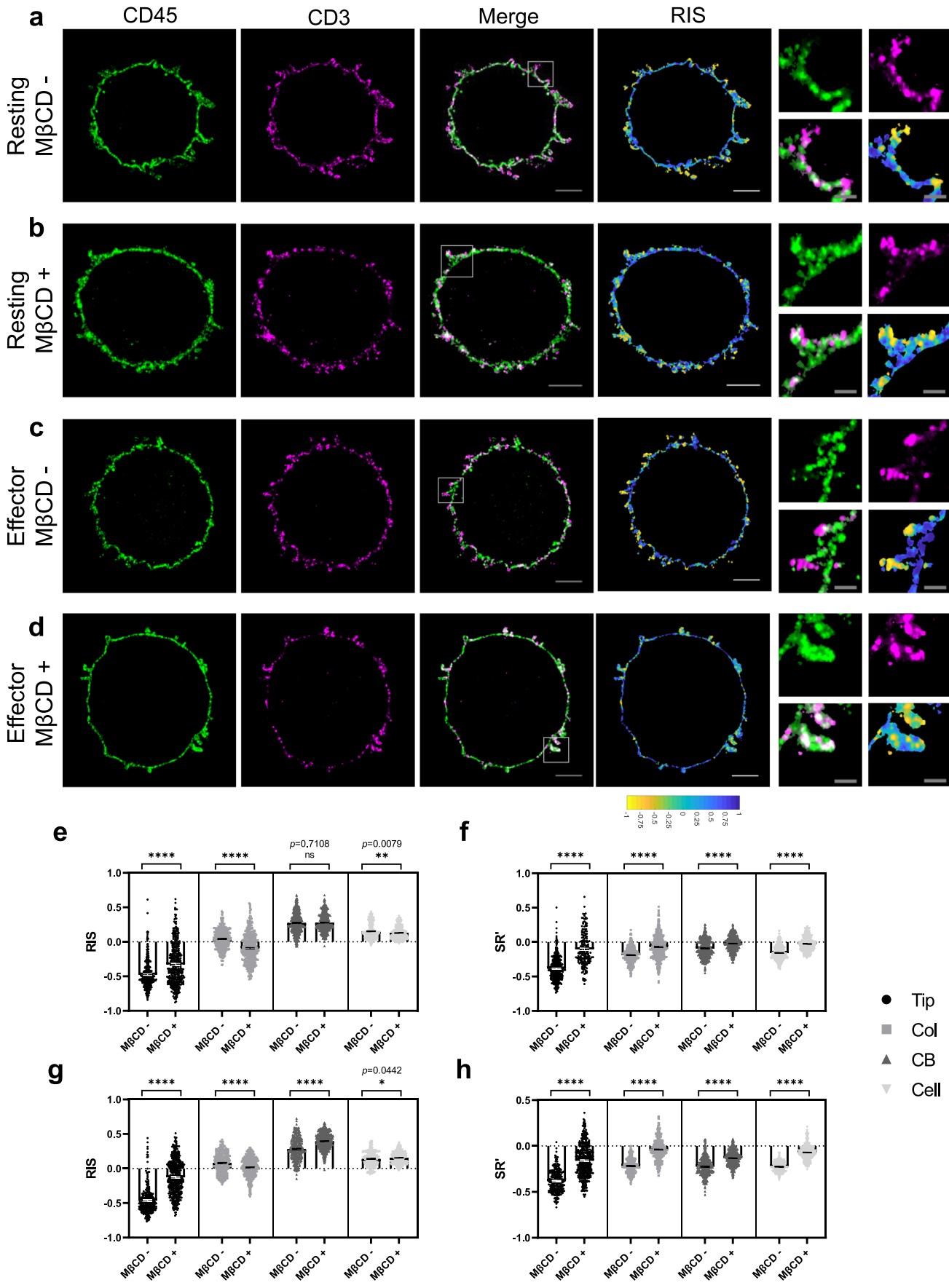

**Fig. 4 Effect of MβCD on the distribution of surface CD45 molecules in human CD4+ T cells. a–d** Representative 4x-ExM-Airyscan images of CD45 (green), CD3 (magenta), the merged, and RIS of representative untreated (MβCD –) or MβCD-treated (MβCD +, 10 mM) resting (**a**, **b**, respectively) and effector (**c**, **d**, respectively) T cells. Magnified images of the area marked in the merged images are presented in the right most panels. Scale bars: 2 μm in the merge and the RIS images; 500 nm in the magnified images. Images in **a–d** are representative of three independent experiments. **e–f** The median RIS values (**e**) and the mean SR′ values (**f**) within the segmented areas, (MV-tip area (Tip, black circle), MV-col area (Col, gray square), CB area (CB, dark gray upward-triangle), or entire cell (Cell, light gray downward-triangle)) were analyzed from each z-plane images (20 z-plane images per cell) of the resting T cells (MβCD –: 19 cells; MβCD +: 21 cells). Each dot represents data collected from a z-plane image. Bars represent the mean and error bars represent the SE. **g**, **h** The median RIS values (**g**) and the mean SR′ values (**h**) of effector T cells (MβCD –: 14 cells; MβCD +: 22 cells) were similarly analyzed as **e**, **f**. *p* values (ns (not significant), *p* > 0.05; *\*p* ≤ 0.05; \*\**p* ≤ 0.01; \*\*\*\**p* ≤ 0.0001) were calculated by two-tailed Mann–Whitney test. Source data for **e–h** are provided as a Source Data file.

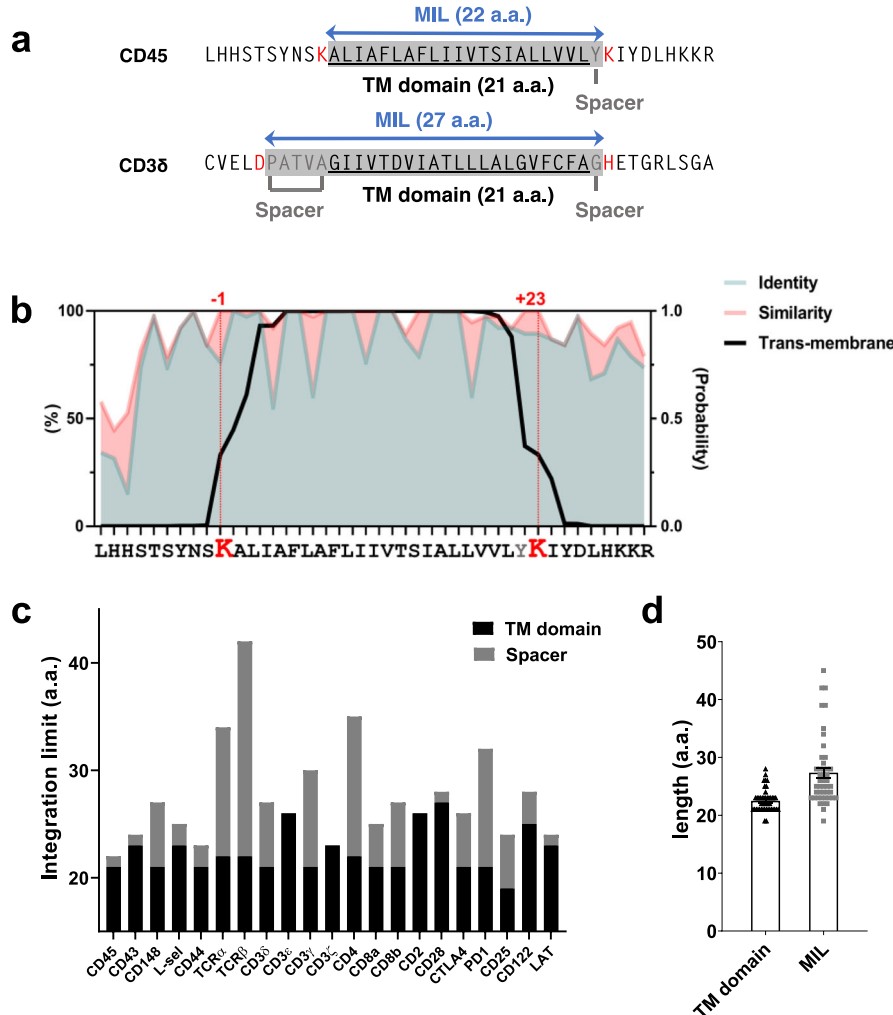

**Fig. 5 Analysis of TM domain and MIL lengths. a** Examples of the MIL of human CD45 and CD3δ. Shown are the a.a. sequences of TM domain (underlined) ±10 flanking residues. The charged a.a. residues located at each end of the TM domain are marked in red; spacers (residues between the TM domain and the charged a.a. residues) are marked in gray; The MIL are marked by blue double arrows. **b** The percentages of sequence identity or similarity (allowing substitutions within the same group of a.a.: charged [K, H, R, E, D]; hydrophobic [A, I, L, F, V, P, G]; polar [Q, N, S, T, C]; amphipathic [W, Y, M]), and the probability of TM positioning of residues analyzed using TMHMM v2.0 (transmembrane helices based on a hidden Markov model[88] (http://www.cbs.dtu.dk/services/TMHMM/) for each residue of the CD45 TM domain ± 10 flanking residues among 38 species (Supplementary Table 3). **c** Lengths of the TM domains (black bars) and spacers (gray bars) of the T cell membrane proteins listed in Supplementary Table 2. **d** Mean lengths of the TM domains and the MILs of non-CD45 membrane proteins (*n* = 45) listed in Supplementary Table 2. Error bars represent the SE. Source data for **b–d** are provided as a Source Data file.

analysis of lipid composition of the cell membrane using nanoscale secondary ion mass spectrometry (nanoSIMS) imaging showed that cholesterol-enriched sphingolipid patches highly accumulated at substrate-bound cellular projections that coincide with the tips of filopodial protrusions[79]. This finding supports our

hypothesis that cholesterol-rich domains are preferentially localized at the tips of MV.

We first tested our hypothesis by using MβCD to deplete membrane cholesterol in resting and effector T cells, and found that this treatment reduced the exclusion of CD45 from the MV

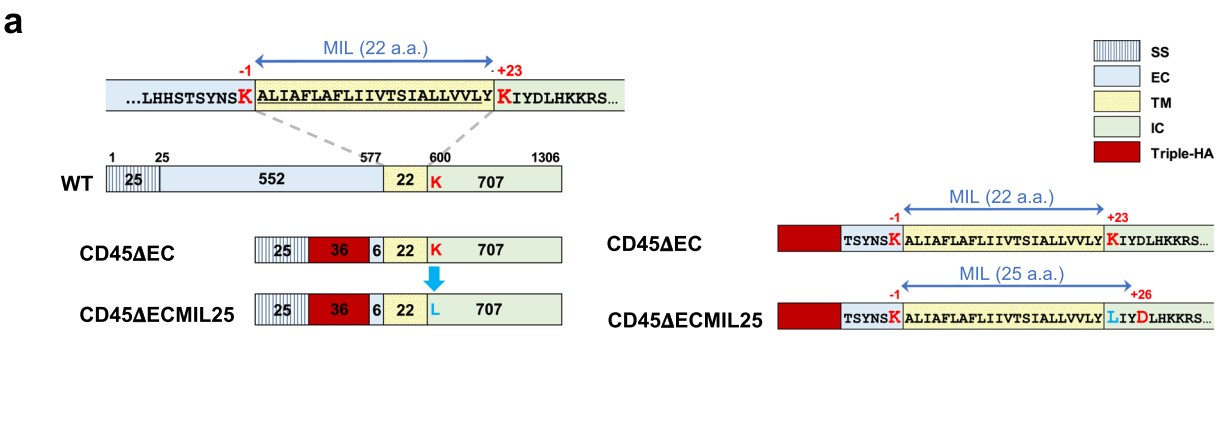

tips. We also observed that MV in MβCD-treated cells were fewer and thinner compared to the ones in untreated cells, especially in resting CD4$^+$ T cells. A similar reduction in the number of MV was reported in MβCD-treated MDCK cells and B cells[55,80], indicating that cholesterol promotes the stability of MV. Secondly, using Jurkat T cells lentivirally transduced with two different CD45 mutants, we ruled out a role for the extracellular CD45 domain in its pre-exclusion. It is consistent with a recent report that the TM domain of CD45 lacking most of its extracellular and intracellular domains segregated out of ordered (raft) domains in the membrane, whereas BCRs were accumulated in

the same domains[81]. Importantly, we demonstrated that increasing the MIL of the CD45 TM domain by replacing the lysine (K) residue in the +23 position with leucine (L) enabled the CD45ΔECMIL25 mutant to diffuse into the MV tips.

The preexisting CD45 exclusion from MV tips, which we discovered here, potentially supports a paradigm of MV serving as sensors for effective antigen recognition[23]. Small phosphatase-depleted zones at the tips of MV that initiate TCR triggering lead to engagement of very few antigen-pMHC complexes while scanning the surface of an APC. Although CD45 is depleted from the tips, the close proximity of CD45 localized on the column are

**Fig. 6 CD45 distribution in Jurkat T cells lentivirally transduced with CD45 mutants. a** A schematic representation of wild-type (WT) human CD45 gene and CD45 mutants, CD45ΔEC and CD45ΔECMIL25. CD45ΔEC (native TM domain) and CD45ΔECMIL25 (K600L) include the signaling sequence domain (SS), a short extracellular domain (six a.a.) (EC), TM domain (TM), intracellular domain (IC), and triple HA tag sequences. MIL of CD45ΔEC and CD45ΔECMIL25 are marked on the right with blue double arrows. **b, c** Representative 4x-ExM-Airyscan images of a Jurkat T cell lentivirally transduced with CD45ΔEC (**b**) or CD45ΔECMIL25 (**c**). CD45 mutants selected among 15 or 25 cells, respectively, observed in three independent experiments. Images of endogenous (green, upper left) or the transduced (magenta, lower left) CD45 mutants, and the merged images between the two are presented. Scale bars: 5 μm. Six z-stack images (step size 125 nm) of the areas marked (1) and (2) in the merged images were magnified in the right panels. Scale bars: 500 nm. The color bar at the bottom represents RIS values between −1 (100% mutant; 0% endogenous CD45) and +1 (100% endogenous CD45; 0% mutant). **d** Colocalization analysis between the endogenous and mutant CD45 of 120 MV images (Supplementary Fig. 11) collected from each CD45ΔEC (black upward-triangle)- and CD45ΔECMIL25 (gray downward-triangle)-transduced Jurkat T cells. Each dot represents data collected from a MV image. Bars represent the mean and error bars represent the SE. Pearson's correlation coefficient (R′, Eq. (2)), Manders' overlap coefficient (MOC, Eq. (4)), Manders' correlation coefficients (MCC1 and MCC2, Eq. (5)) were calculated as described in Supplementary Table 1 and Methods. $p$ values (****$p < 0.0001$) were calculated by two-tailed Mann–Whitney test. Source data for **d** are provided as a Source Data file.

able to access the intracellular position of Lck (which is also expressed on MV[77]) that requires for positive signaling[8,9]. It may lead MV collapsing mediated by Rac1 and ERM family[77,82] and actin-remodeling[83]. These leads flattening of the interface that drives out CD45 from the IS by the size exclusion mechanism. Finally, a mature IS is formed to fully establish a productive T cell response[84].

## Methods

**Mice**. C57BL/6 J mice (male or female, 11–33 weeks old) were purchased from Jackson Laboratory. Foxp3[YFP-Cre] reporter mice (B6.129(Cg)-Foxp3[tm4(YFP/icre)Ayr]/J, Jackson Laboratory, male, 12–13 weeks old) were kindly gifted by Dr. Lynn Hedrick (La Jolla Institute for Immunology). OT-II x Rag2[−/−] mice (male or female, 11–31 weeks old) were generated by crossing male OT-II x Rag2[−/−] mice (B6.129S6-Rag2[tm1Fwa] Tg(TcraTcrb)425Cbn, Taconic) kindly gifted by Dr. Stephen Schoenberger (La Jolla Institute for Immunology) with female Rag2[−/−] (B6.129S6-Rag2[tm1Fwa] N12, Taconic). Temperature in the vivarium was maintained at an average of 72 °F (range = 69–75 °F), with an acceptable humidity range of 30–70%. Cycles of light followed by cycles of darkness (12 h each) controlled automatically via building management system. Mice were maintained and used by following the guidelines of the La Jolla Institute for Immunology animal care and use committee, and approval for use of mice was obtained from the La Jolla Institute for Immunology according to criteria outlined in the NIH guide for the care and use of laboratory animals.

**Cells**. Human peripheral blood mononuclear cells (PBMCs) were prepared from the whole blood of healthy donors by the La Jolla Institute for Immunology clinical core via La Jolla Institute normal blood donor program (VD-056) approved by the La Jolla Institute for Allergy & Immunology IRB #1 (IRB registration number: IRB0000850; federalwide assurance number: FWA00000032). Human CD4[+], CD8[+], and B cells were purified from PBMCs using negative selection kits, EasySep™ Human CD4[+] T Cell Isolation Kit, EasySep™ Human CD8[+] T Cell Isolation Kit, or EasySep™ Human B Cell Isolation Kit (StemCell Technologies). Freshly isolated resting cells were used after incubation in complete phenol red-free RPMI (Gibco) growth medium [2 mM L-glutamine (Gibco), 1 mM sodium pyruvate (Gibco), 1 mM nonessential amino acids (Gibco), 100 U penicillin-streptomycin (Gibco), and 10% (v/v) FBS (Gemini Bio)] for ~2–4 h in a CO₂ (5%) incubator at 37 °C. For effector T cells, freshly isolated human CD4[+] T cells were stimulated on an anti-CD3 (OKT3, Biolegend) plus anti-CD28 (CD28.2, Biolgend) antibody-coated plate for 2 days and then transferred and grown in a complete medium supplemented with IL-2 (100–130 U mL[−1]) and 2-mercaptoethanol (50 μM) for 7 days. Mouse spleens and lymph nodes were processed through a 70 μm filter (BD Biosciences) and the single cell suspensions were washed and incubated with RBC lysis buffer (eBioscience) for 5 min at room temperature (RT). After washing, CD4[+] T cells or B cells were purified using EasySep™ Mouse CD4[+] T Cell Isolation Kit or EasySep™ Mouse B Cell Isolation Kit (StemCell Technologies). The purified cells were used after a short incubation at 37 °C as described above for human cells. For ODT live imaging, OVA-specific CD4[+] T cells were isolated from spleens and lymph nodes of TCR-transgenic OT-II mice that were crossed to Rag2[−/−] mice, and mouse B cells used as APCs were isolated from spleen and lymph nodes of gender and age matched C57BL/6 J mice using the EasySep negative selection kits. Cells were incubated in complete RPMI growth medium for more than 1 h in a CO₂ incubator at 37 °C after the purification. Jurkat T cells (Clone E6-1, gift from Dr. Zachary Katz, La Jolla Institute for Immunology) were grown in phenol red-free RPMI growth media containing 2 mM L-glutamine, 100 U penicillin-streptomycin, and 10% FBS. The 293T cells were grown in DMEM medium (Invitrogen) containing 10% FBS and 100 U penicillin-streptomycin.

**Antibodies**. Mouse anti-human CD62L (L-sel; DREG-56, Biolegend) were labeled with AF568 using an AF568 antibody labeling kit (Thermo Fisher). Purified mouse monoclonal antibodies specific for human CD3 (UCHT1, Biolegend), HA.11 epitope tag (16B12, Biolegend), human IgM (MHM-88, Biolegend), and human IgD (IA6-2, Biolegend) or rat anti-mouse CD62L (MEL-14, Biolegend) were labeled with CF633 using Mix-n-Stain™ CF™ 633 antibody labeling kit (Sigma). The following labeled primary antibodies from commercial sources were used: Alexa Fluor® 488 anti-human CD45 Antibody (Biolegend, Cat# 304017, clone HI30), Alexa Fluor® 647 anti-human CD45 Antibody(Biolegend, Cat# 304056, clone HI30), Alexa Fluor® 488 anti-mouse CD45 Antibody (Biolegend, Cat# 103122, clone 30-F11), LEAF™ Purified anti-human CD62L Antibody (Biolegend, Cat# 304812, clone DREG-56), LEAF™ Purified anti-mouse CD62L Antibody (Biolegend, Cat# 104416, clone MEL-14), LEAF™ Purified anti-human CD3 Antibody (Biolegend, Cat# 300414, clone UCHT1), Purified anti-human IgM Antibody (Biolegend, Cat# 314502, clone MHM-88), Purified anti-human IgD Antibody (Biolegend, Cat# 348202, clone IA6-2), Ultra-LEAF™ Purified anti-HA.11 Epitope Tag Antibody (Biolegend, Cat# 901521, clone 16B12), Ultra-LEAF™ Purified anti-human CD3 Antibody (Biolegend, Cat# 317326, clone OKT3), Ultra-LEAF™ Purified anti-human CD28 Antibody (Biolegend, Cat# 302934, clone CD28.2). For cell surface labeling, 10 μg of antibodies per 1 × 10[6] cells in 100 μl volume were used (1: 40–1: 200 dilution).

**Sample preparation for imaging**. Purified human cells suspended in cold PBS (Gibco) containing 5 mM EDTA (Sigma) were washed by centrifugation at 4 °C, resuspended in blocking buffer [PBS containing 2% BSA, 5 mM EDTA, 10 mM EGTA, and 0.05% N₃Na], and incubated on ice for 10 min. Fluorescently labeled antibodies (10 μg ml[−1]) were added to the cell suspension and incubated for additional 20 min on ice. Cells were washed 2x with 5 mM EDTA/PBS and fixed with fixation buffer [4% (w/v) paraformaldehyde (PFA) and 0.5% (w/v) glutaraldehyde (GA; both from Electron Microscopy Sciences), 2% (w/v) sucrose, 10 mM EGTA, 5 mM EDTA, and 0.05% (w/v) N₃Na (Sigma) in PBS] on ice for 1–2 h. Cells were washed 2x with PBS, resuspended in cold PBS, and kept on ice. For staining antibodies (CD45-AF488 and L-sel-AF568) at 37 °C samples in the Supplementary Fig. 2, cells were spin-washed once with pre-warmed 2% FBS/PBS or cytobuffer [2 mM MgCl₂, 5 mM EGTA, and 2% sucrose in 80 mM HEPE) and mixed with antibodies (10 ug ml[−1] for each) and incubated at 37 °C for 10–15 min. Cells were washed once with 2% FBS/PBS or cytobuffer and fixed with at RT as described earlier. For MβCD treatment, cells were washed 2x with serum-free RPMI at RT and incubated without or with 10 mM of MβCD (Sigma) in RPMI containing 25 mM HEPES (Sigma) in a CO₂ incubator for 30 min at 37 °C. Cells were washed 2x with cold 5 mM EDTA/PBS at 4 °C, and then incubated in blocking buffer containing 2% (w/v) fatty acid-free BSA (Sigma), 5 mM EDTA, 10 mM EGTA, and 0.05% N₃Na in PBS. For mouse samples, mouse T cells and B cells were prepared as described above except cells were resuspended, washed, and fixed in cytobuffer. For membrane labeling, the cells were stained with fixable membrane dye, FM 4-64FX (Invitrogen) after overnight incubation for the 4x-ExM sample preparation (see the 4x-ExM-Airyscan microscopy section). After washing with PBS at RT, cells were stained with FM 4-64FX (5 μg ml[−1]) in PBS for 1 h at RT. Cells were washed 3x with PBS, and then fixed and washed as described above at RT for 1 h. For ODT live imaging, isolated OT-II T cells were stained with Fluo-4M (2.5 μM; Molecular Probes) and incubated in a CO₂ incubator at 37 °C for 30 min. C57BL/6 J mouse B cells, used as APCs, were washed and resuspended in serum-free, phenol red-free RPMI and were incubated with 10 μg ml[−1] of OVA₃₂₃₋₃₃₉ peptide (Invivogen) at 37 °C for 1 h. After 30 min, Cell Tracker Orange CMRA Dye (0.5 μM; Molecular Probes) was added to the cells. Both OT-II T cells and B cells were washed twice and resuspended in imaging buffer [48% phenol red-free RPMI, 50% HBSS (Gibco),and 2% FBS].

**Plasmids and lentiviral transduction**. DNA sequences encoding the CD45 mutants, CD45ΔEC and CD45ΔECMIL25, were synthesized (Integrated DNA

Technologies) and cloned into the lentiviral vector pFUW[85] (gift from Alok Joglekar and David Baltimore). The resulting vectors pFUW-CD45ΔEC or pFUW-CD45ΔECMIL25 were confirmed by DNA sequencing. The two vectors along with two packaging plasmids pSPAX2 (Addgene) and pMD2.G (Addgene) were cotransfected into 293T cells using the TransIT-LT1 Transfection Reagent (Mirus Bio LLC) according to the manufacturer's instructions. The supernatant was collected after 48 h, and applied to Jurkat T cells for viral delivery. Two days later, Jurkat T cells were harvested for microscopic or for flow cytometric analysis.

**Flow cytometry and sorting.** Some of the MβCD-treated cells were collected from the blocking reaction to determine viability by diluting them with PBS and staining with 7-AAD (20 µg ml$^{-1}$, BD Biosciences) for 20 min on ice, followed by washing with cold 1% FBS/PBS. The cells were resuspended in 200 µl of 1% FBS/PBS and kept on ice in the dark. Cells were immediately loaded for flow cytometry acquisition on a BD FACSCelesta (BD Biosciences). Single cells gated using forward vs. side scatter (FSC vs. SSC) were analyzed for 7-AAD fluorescence using FACSDiva software (BD Biosciences, version 8.0.1). Data analysis were done with FlowJo (BD Biosciences, versions 10.4.2 and 10.6.2) software. For sorting the Treg cells, CD4$^+$ cells isolated from *Foxp3*$^{YFP-Cre}$ mice were labeled with Alexa Flour 488-conjugated anti-mouse CD45 and CF633-conjugated anti-mouse L-sel on ice for 20 min as described in sample preparation for mouse cells. Fixed cells were resuspended in 0.5 ml PBS and were kept at 4 °C until sorting. After selecting single cells determined by FSC vs. SSC gating, CD45$^+$Lsel$^+$YFP$^+$ cells were sorted and collected in PBS on a BD Aria III or Fusion cell sorter (BD Biosciences). Jurkat T cells transduced with CD45 mutants CD45ΔEC and CD45ΔECMIL25 were washed 2x in serum-free RPMI at RT. Alexa Flour 488-conjugated anti-human CD45 and CF633-conjugated anti-HA were labeled in 2% FBS/PBS for 20 min at RT. After washing twice, cells were fixed with 4% (w/v) PFA, 0.5% GA, 2% sucrose, 10 mM EGTA, 5 mM EDTA, and 0.05% N$_3$Na in PBS on ice for 1 h at RT. Cells were washed twice in PBS and resuspended in 0.5 ml PBS. After selecting single cells determined by FSC vs. SSC gating, CD45$^+$ cells expressing high level of HA were sorted and collected in PBS on a BD Aria III or Fusion cell sorter (BD Biosciences). The sorted cells were processed for 4x-ExM-Airyscan imaging as described below.

**4x-ExM-Airyscan imaging.** The 4x-ExM samples were prepared as described in refs. [40,41] with some modifications. Succinimidyl ester of 6-((acryloyl) amino)) hexanoic acid (0.1 mg ml$^{-1}$; acryloyl-X, SE, Life Technologies) was added to the fixed cells in PBS and then the 35 µl of cells were placed on freshly prepared poly-L-lysine (PLL; 0.1%; Sigma)-coated 5 mm round glass coverslips (Electron Microscopy Sciences). Samples were kept overnight at 4 °C, and rinsed 4x with PBS at RT. Buffer was removed by gentle suction and then the cell-coated coverslip was quickly placed on top of a second rectangular glass coverslip (15 mm × 15 mm; Electron Microscopy Sciences) that had a drop of freshly mixed gelation solution. The gelation solution was freshly mixed with monomer solution [PBS supplemented with 2 M NaCl, 8.625% (w/w) sodium acrylate, 2.5% (w/w) acrylamide, 0.15% (w/w) N, N'-methylenebisacrylamide, ammonium persulfate initiator (APS; 0.2%; Sigma), and tetramethylethylenediamine (TEMED; 0.2%; Sigma)], kept on ice before use. Gelation samples were sealed and kept in the dark for 45 min at RT, followed by digestion of the gel in digestion solution [50 mM Tris buffer (pH 8.0) containing 1 mM EDTA, 0.5% Triton X-100, 0.8 M guanidine HCl (Sigma), and 1% (v/v) proteinase K (New England Biolabs)] for 2 h at 37 °C with gentle shaking. Gels formed between the two coverslips were detached during the digestion, fully expanded with an excess of H$_2$O for 30 min at RT, and immobilized on PLL-coated (0.1%; Sigma) coverslip for microscopy. Cells were placed on a laser scanning confocal microscope ZEISS LSM 880 equipped with an Airyscan detector (Carl Zeiss). Airyscan images were taken with a C-Apochromat 40x/1.2 W AutoCorr M27 objective with a 100–130 µm sized pinhole with master gain 850 or 950 while keeping humidity in the sample chamber by a humidifier. The 488-, 561-, or 633-nm laser lines (Carl Zeiss) were used to excite the AF488-, AF568-, and CF633-labeled samples, respectively, and a 561-nm laser line was used to excite FM 4-64FX. Doubly labeled samples (AF488- and CF633- or AF568-conjugated antibodies) were acquired with interleaved laser excitation (ILEX) mode. The excitation and emission for each channel were separated with a combination of a filter set of a beam splitter MBS 488/561/633 (Carl Zeiss) and multi bandpass filter BP 495–550 + LP 570 (Carl Zeiss). Triple-channel images were acquired sequentially with the main beam splitter MBS 488/561/633 combined with SBS SP 615 (Carl Zeiss) and BP 420–480 + BP 495–550 for AF488-labeled samples, SBS SP 615 and BP 495–550 + LP 570 (Carl Zeiss) for AF568-labeled samples; and SBS LP 660 (Carl Zeiss) and BP 495–550 + LP 570 (Carl Zeiss) for CF633-labeled samples and FM 4-64FX-labeled samples. For each channel, ~80–150 series of z-plane Airyscan confocal images were taken with 0.25 µm steps. The x and y step sizes (pixel size) were 50 nm. ZEN Black 2.3 SP1 FP2 software (Version 14.0.16.201) was used for the post-3D Airyscan processing with automatically determined default Airyscan Filtering (AF) strength and for the 3D reconstruction images.

**STORM imaging and analysis.** For STORM imaging, µ-Slide 8-well glass bottom chambered coverslips (Ibidi) were cleaned with 1 M NaOH (Sigma) for 1 h, and coated with PLL (0.1%) for 1 h at RT before use. Fixed cells labeled with AF647- or AF568-conjugated antibodies and suspended in 300 µl PBS were placed in each well

incubated for 1 h at RT, and then the buffer was exchanged with blinking buffer [PBS containing 0.5 mg mL$^{-1}$ glucose oxidase, 40 µg mL$^{-1}$ catalase, 10% (w/v) glucose, 50 mM cysteamine, and 93 mM Tris-HCl (all from Sigma)]. All buffers were filtered with 0.2 µm-pore filters (Corning) and freshly mixed before use. For fiducials, Fluorescent Nanodiamond (FND) (100 nm, Adamas Nano) diluted in water (1:1000) was sonicated for 15 min and then added to the PLL-coated wells for 30 min. Unbound FNDs were washed 5x with distilled water and then blinking buffer was added. STORM imaging was acquired with a UAPON100XO-TIRF1.49NA oil objective (Olympus) at epi-illumination mode on Nanoimager (ONI) microscope implemented with a cylindrical lens for localization in z-dimension. The focus was maintained by the autofocus module of the system. Excitation lasers 532 and 640 nm and the two channels were separated with dichroic (640 LP) and emission (584/80 and 685/40) filters. 532 and 640 nm laser lines were operating at a power of 27.5 and 13.8 kW cm$^{-2}$, respectively. For each channel, 30,000 frames (3000 frames x 10 movies) were recorded with a speed of 10 ms per frame. The pixel size on the detector was 117 nm. The two-channel registration with FND fiducial images in the two-channel and single molecule localization processing was performed using Nanoimager software (version 1.1.6165-012f4ed3; ONI). Data analysis and reconstruction of images were performed with custom-written Matlab (MathWorks, 2018a) scripts. Single molecule localization data were thresholded as: 0–14 nm for "precision in x, y (nm)"; −400 to 400 nm for "z range"; 20,000 for "number of photons"; 100 for "background"; 0–2.5 for "Point spread function σX and σY (pixel)'; 0.6–1.5 for "σX/σY" (pixel)'. Channel correction for individual cell images was done with low-resolution images of the single molecules collected in all frames rendered with the original pixel size by calculating the shifted peak of the 2D cross-correlation image obtained using a 2D Gaussian fit. The tip positions were determined using the 2D rendered image (ignoring z localization) as described in the area segmentation section below. Molecules that are localized within the 9 × 9 pixels (corresponding to 1.053 µm × 1.053 µm) around the tip positions were analyzed for the quantification. 3D-pair-correlation (g(r)) between the L-sel and CD45 was calculated. Briefly, the mean count of all CD45 molecules localized within a distance between $r$ (radii) and $r + dr$ ($dr = 10$ nm) away from each L-sel molecules was calculated, and then it was divided by the volume of the spherical shell. Finally, this number was divided by the total density of the CD45 molecules.

**Optical diffraction tomographic microscopy.** Freshly prepared OT-II T cells loaded with Ca$^{2+}$ indicator Flou-4-AM and antigen-pulsed B cell APCs labeled with Cell Tracker Orange CMRA Dye were mixed 1:1, and were then immediately loaded onto a TomoDish (Tomocube). Samples on TomoDish were placed on a TomoChamber (Tomocube) to maintain CO$_2$ (5%) and temperature (37 °C) levels. Cells were subjected to time-lapse imaging with an objective and a condenser lens, UPLASAPO 60XW 1.2NA lens (Olympus), on a holotomographic microscope, HT-2 (Tomocube). Holographic images were generated from interfered images at the camera plane between the two split 532 nm laser beams of a reference beam and a sample illuminated beam obtained at various incident angles modulated by a high-speed illumination scanner using a digital micromirror device (DMD). Following the holographic imaging acquisition in 400 ms, single z-plane fluorescent images of Fluo-4 AM for T cells and the Cell Tracker Orange CMRA for B cells illuminated with an LED light source (470 and 570 nm, respectively) were sequentially illuminated with 100 ms exposure time. Time-lapse movies were taken for 5 min with 11 s intervals for each field of view (37 µm × 37 µm). The refractive index (RI) distribution was reconstructed and visualized for the 3D ODT images using the Tomostudio software (Tomocube, version HT-2H-2.7.35). The trace of Fluo-4 AM fluorescence intensities for each frame of the time-lapse movie was analyzed using a custom-written Matlab script. Briefly, Fluo-4 AM channel image was binarized using the function "imbinarize" (Matlab) applying "global" image threshold (using Otsu's method). Holes in the images were filled with the "imfill" function (Matlab, 2018a). The region of interest (ROI) of a T cell area is determined as of the biggest connected component (with the connectivity of 8) found in the binary image using the "bwconncomp" function (Matlab). The mean intensities of the Fluo-4 AM within the ROI were calculated for each frame.

**Area segmentation.** The MV tips, MV-tip area, MV-col area, and CB area of the 4x-ExM-Airyscan images were segmented using a custom-written Matlab (Math-Works, R2018a,) script. Firstly, to exclude background signals from the outside-cell area (OutBG, non-cell area) or from the inside-cell area (InBG, such as nucleus region), the OutBG and InBG were segmented as follows: each image channel was converted to binary images (BW1 for channel 1 and BW2 for channel 2) applying the first threshold value obtained using the function "multithresh" in Matlab. The summed image of the two-channel images was converted to a binary image with a threshold obtained by using the "imbinarize" function in Matlab with an "adaptive" option with the "sensitivity" parameter set to 1. Holes in the binary image were filled with the "imfill" function in Matlab. The OutBG was the area except for the biggest connected component (with the connectivity of 8) found in the binary image using the "bwconncomp" function in Matlab. The InBG area was segmented by "activecontour" function (applying the "edge" method and 25 iterations) in Matlab. Seed mask for the "activecontour" function was generated from dilated images of OutBG. The InBG area was finally obtained after dilated with the function "imdilate" with a disk structuring element with a radius of 5 pixels and holes were filled by "imfill" function. Each channel images were converted to binary

images using the "graythresh" function with "global" image threshold (using Otsu's method) and two binary images were added and the non-zero elements of this image were converted to a binary image, and then remove the OutBG or the InBG areas in this binary image (SegBW). SegBW is the area of entire cell area (Cell) for further analysis. Next, the SegBW was segmented to microvilli area (MV-BW) or cell body area (CB-BW). The CB-BW was defined by the SegBW area overlapped with the dilated area from the InBG mask image using the "imdilate" function with a disk structuring element with a radius of 24 pixels (300 nm), and the rest of the area in the SegBW was determined as MV-BW. The locations of MV tips were determined as follows: The components in the SegBW were connected using imclose function with a disk structuring element with a radius of 5 pixels and then, the small-sized (<2000 pixels) unconnected components were removed for eliminating disconnected MV from a cell. The binary image of the MV area in the SegBW was processed with "imclose" ("disk", element radius set to 10) and "imerode" ("cube", element pixels set to 5) functions, and then were applied to the "bwconvhull" function (Convex hull algorithm) in Matlab. The output a convex hull image from the "bwconvhull" function was eroded with the "imerode" ("cube", element pixels set to 5) function and the zero-elements of this convex hull image overlapped with the MV area were selected for generating a binary image for tips' areas (TipBW). The TipBW were dilated with the function "imdilate" ("cube", element pixels set to 3) and individual tip-areas (L-BW) were determined with "bwlabel" function with the parameter for connected object size set to 8. The dilated TipBW image was eroded back with the function "imerode" ("cube", element pixels set to 3) and the mean x,y locations of individual L-BW pixels overlapped with the TipBW were calculated for the location for MV tips. The length of MV was calculated from the closest distance of the location of the MV tip to an inner boundary of cell determined from the outer-edge pixels of InBG binary image using the "edge" function with "Sobel" method with threshold parameter set to 0 and skipping the edge-thinning stage. Any MV tip location for MV whose length was shorter than 50 nm or larger than 1.5 μm in human and mouse T cells were excluded for calculation. If any two locations of the MV tip were closer than 50 nm, the MV tip location for the shorter MV was excluded for the calculation. The distance from tips of MV was calculated using the function "pdist2" in Matlab. Finally, the MV-BW was divided to MV-tip area or MV-col area as follow: MV-tip area was determined where the pixel distances were less than 150 nm from each MV tips and the rest of the MV-BW was determined as MV-col area. The junctional regions (75 nm) between MV-tip area and MV-col area were removed from MV-col area (gray-color in segmented map images). The CB area was determined from CB-BW after removing the junctional areas between the MV-tip area and CB-BW.

**Image analysis**. RIS (Eq. (1)), Pearson's correlation coefficient (R′, Eq. (2)), segment-correlation coefficient (SR′, Eq. (3)), Manders' overlap coefficient (MOC, Eq. (4)), and Manders' correlation coefficients 1 and 2 ($MCC_1$ and $MCC_2$, Eq. (5)) values were calculated as shown in Supplementary Table 1. RIS and SR′ calculations were performed with data within the SegBW. For intensity distribution plots in Figs. 2g, i, 3g, i and intensities of the segmented areas such as Fig. 2h, j, intensities of each channels were normalized to 0–1 value after saturating the upper 1% of the intensity after background subtraction. The background threshold values were determined using the "graythresh" function in Matlab. For RIS and SR′ calculation, original intensities were used without subtracting background. For colocalization analysis in Fig. 6d, the first threshold values obtained using the "multithresh" function in Matlab were used for thresholding the positive intensities for each channel. The display range of the intensities of individual color channels in Fig. 1, Supplementary Figs. 1, 6, and Supplementary Movie 1 were independently optimized using Zen Black program (Carl Zeiss). ROIs were selected by Image J (version 1.52p), Zen Black program (Carl Zeiss), or Matlab (MathWorks, R2018a). The display ranges of intensities in Supplementary Fig. 3c, d (L-sel-low) for the L-sel channel were adjusted to that of Supplementary Fig. 3a for comparison. All the other 4x-ExM-Airyscan images in this study were displayed between the 1–99.5% intensity of individual channel image using a built-in function in Matlab.

**Statistical information**. Statistical details of experiments can be found in the figure legends and exact p values are indicated in the figures when P > 0.0001. Mean, median, standard deviation (SD), and standard error of the mean (SE) were determined using the built-in functions in Matlab (MathWorks, R2018a) or Prism 8 and 9, (GraphPad, version 8.4.1; version 9.0.1). Two-tailed Wilcoxon matched-pairs signed rank test and two-tailed Mann–Whitney test were performed using Prism 8 and 9 (GraphPad) software.

**Reporting Summary**. Further information on research design is available in the Nature Research Reporting Summary linked to this article.

## Data availability
Imaging data that are generated and used in this study are available from the corresponding author upon reasonable request. The sequences and the predicted lengths of TM domains listed in Supplementary Table 2 and 3 are available in in the Universal Protein Resource (UniProt; https://www.uniprot.org/) or reported in refs. [86,87] The probability of TM domain of CD45 in Fig. 5b was analyzed using TMHMM v2.0

(transmembrane helices based on a hidden Markov model[88] (http://www.cbs.dtu.dk/services/TMHMM/). Source data are provided with this paper.

## Code availability
Custom codes used in this study are available from GitHub (https://github.com/ymjung1/ncomms_2021_Jung).

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

## Acknowledgements

This is manuscript number 3383 from the La Jolla Institute for Immunology (LJI). We thank Drs. Sara McArdle (La Jolla Institute for Immunology), Enrico Gratton (University of California, Irvine), and Gilad Haran (Weizmann Institute of Science) for helpful discussions on imaging analysis, Dr. Padmini Rangamani and Mr. Arijit Mahapatra (UCSD) for discussions on membrane modeling, Ms. Kristine Suchey (Division of Laboratory Animal Care, LJI) for collecting mice tissues, and members of the microscopy and flow cytometry cores (LJI) for technical support. We appreciate CTK Instruments and Mr. Adam Lucio for offering us to use the Tomocube system and for the technical support. This research was supported by NIH grants S10OD021831 and P01 HL078784 project 3 to K.L.

## Author contributions

Conceptualization, Software, Formal analysis, Investigation, Data curation, Writing—original draft, and Visualization, Y.J.; Methodology, Y.J. and L.W.; Resources, K.L.; Writing—review and editing, Y.J., A.A., and K.L.; Supervision, K.L. and A.A.; Funding acquisition, K.L.

## Competing interests

The authors declare no competing interests.
