## [Peer Review File · Nature Communications]

REVIEWER COMMENTS

Reviewer #1 (Remarks to the Author):

The authors use expansion microscopy and Airyscan microscopy to achieve 40 nm resolution of protein distributions in small projections on T cells that are widely referred to as microvilli. They show that L-selectin and TCR are biased to microvilli tips as previously described using TIRF imaging by some the first author. The authors here also show this by dSTORM. The authors convincingly show that under the experimental conditions CD45 is excluded from the microvilli tips and that normal levels of plasma membrane cholesterol are critical for the CD45 exclusion from the tips, but that the CD45 extracellular domain is not required. This results is different from the TIRF analysis from Haran's lab from 2016 and a more recent paper from Haran lab what argues that CD45 distribution seemed to be uniform. There are potential issues with the conditions that need to be considered (see below). The data on Calcium imaging and proof that the Calcium flux is induced by microvilli contacts is not convincing as presented, but with controls and further analysis this seems like a productive direction. The working model presented is quite confusing as they start with a one-sided presentation of kinetic segregation as a model to explain TCR triggering and then conclude that CD45 is already segregated from the TCR at microvilli tips, which if true would largely invalidate the simplest version of the K-S model.

Major issues

Fixing the cells after holding them at 4 degrees is a problem for this experiment as work from Baird and colleagues has shown that cooling cells results in generation of microscopic liquid ordered domains that cannot be detected at 37 degrees. See Baumgart T, Hammond AT, Sengupta P, Hess ST, Holowka DA, Baird BA, Webb WW. Large-scale fluid/fluid phase separation of proteins and lipids in giant plasma membrane vesicles. *Proc Natl Acad Sci U S A*. 2007;104(9):3165-70. Epub 2007/03/16. doi: 10.1073/pnas.0611357104. PubMed PMID: 17360623. This has been explicitly demonstrated for T cells- Magee AI, Adler J, Parmryd I. Cold-induced coalescence of T-cell plasma membrane microdomains activates signalling pathways. *J Cell Sci*. 2005;118(Pt 14):3141-51. Epub 2005/07/15. doi: 10.1242/jcs.02442. PubMed PMID: 16014381. It would be much better if the labelling could be done with monovalent, non-triggering/inhibitory reagents at 37 degrees and then the cells fixed at 37 degrees prior to analysis. I suspect that the liquid ordered domains are formed at the tips of microvilli at low temperature, and this probably means something biological based on the Magee et al paper, but I expect that CD45 will not be excluded from microvilli tips at 37 degrees. Certainly, Cai et al 2017 didn't see any evidence of this and neither of the studies from Haran lab, but perhaps the resolution was inadequate in those situations.

The ODT/Calcium measurement approach clearly has promise, but the results are preliminary. Many papers have shown that Calcium flux precedes stable synapse formation since the mid 1990's and maybe even earlier, but this is quite explicitly stated and examples are shown in the Bunnell et al 2002 paper and in the Campi et al 2005 paper. But it has not be well documented that these are contacts with microvilli. It looks like this is going in the correct direction, but just needs better quantification and controls.

Minor issues

Statistical analysis of a parameter related to TCR clustering and CD45 exclusion from the tips would be important. The general polarization value is misnamed as it contains no information about spatial polarization, which would tip/shaft comparison. They could say that they mean "polarization" of the pixel intensity of the parameter, but this seems to be asking for confusion. The parameter is really a "signed segregation" score that goes to 0 when both molecules have similar normalized intensity and is +1 when only X is present and -1 when only Y is present. The spatial information only comes into play when the parameter is plotted against distance from the tip. A general polarization score might be developed by taking a ratio of tip vs column, but this would require a decision as to where to draw this line between these two compartments. The use of the Pearson's correlation coefficient is also help as its not biased by any author decision. But I feel that a parameter that capture the tip/column ratio and enable statistical testing on this conclusion is important.

The introduction is biased toward kinetic segregation model as a primary triggering mechanism. Some of the earliest studies on CD45 knockout T cells showed that function can be restored by the forms of CD45 lacking the extracellular domain (Volarevic S, Niklinska BB, Burns CM, June CH, Weissman AM, Ashwell JD. Regulation of TCR signaling by CD45 lacking transmembrane and extracellular domains. *Science*. 1993;260(5107):541-4. PubMed PMID: 8475386.). The authors discuss a general exclusion of CD45 from the entire cell interface, but application of TIRF to T cell activated on substrates demonstrated that CD45 exclusion was very local to TCR clusters, which are likely collapsed microvilli (Varma R, Campi G, Yokosuka T, Saito T, Dustin ML. T cell receptor-proximal signals are sustained in peripheral microclusters and terminated in the central supramolecular activation cluster. *Immunity*. 2006;25(1):117-27. Epub 2006/07/25. doi: 10.1016/j.immuni.2006.04.010. PubMed PMID: 16860761). Finally, a recent paper using nanofabrication methods demonstrated that preventing CD45 exclusion results in a greater requirement for receptor clustering, but didn't prevent triggering (Cai H, Depoil D, Muller J, Sheetz MP, Dustin ML, Wind SJ. Spatial Control of Biological Ligands on Surfaces Applied to T Cell Activation. *Methods Mol Biol*. 2017;1584:307-31. doi: 10.1007/978-1-4939-6881-7_18. PubMed PMID: 28255709.). There is also the work from Weiss on the importance of CSK. So I think there are a number way to think about triggering. Certainly, if the authors continue to find CD45 exclusion even at 37 degrees then it would be ideal to be less biased, which make it easier to consider how the system could function with CD45 already segregated from the TCR.

Reviewer #2 (Remarks to the Author):

Here, the authors use expansion microscopy to analyse the distribution of CD45 in microvilli at the T cell immunological synapse. It has been widely postulated that CD45 needs to be excluded from sites of TCR engagement in order to facilitate signalling but this is the first work showing that the phosphatase is pre-excluded from the very initial contact sites. Overall, I think this is an important finding and I support publication. The paper is also technically quite sound and so I don't think it needs that much work to bring it to a publishable standard. Overall then I recommend publication subject to the following corrections, which I don't think are too difficult.

1. There is a lack of image quantification and statistical testing, even starting from the very First Figure. The authors should apply a method of colocalization (e.g. Pearson's) to the microvilli tips and sides and show that they are indeed statistically significantly different. That is the case for all claims of differential colocalization throughout the manuscript.
2. To me, the STORM experiments don't add much understanding. If I were them, I'd leave this out, or, if they stay in they need to be quantified by some method of colocalization too.
3. For Fig 2, I don't really understand the motivation for the use of the GP measure – rather than a simple ratio or other method of colocalization such as Mander's? Could the authors explain why they chose this procedure and how statistical significance can be inferred from it?
4. The hypothesis that cholesterol accumulates at the tips seems out of the blue – why did they authors hypothesise this – is there literature on the subject? The use of MbCD has been problematic in some cell types – possibly creating gel phases etc and some, such as the Gaus lab have used 7-ketocholesterol as an alternative for ablating the membrane ordered phase. I think the authors should consider this, but I'd be happy with some discussion on the potential limitations/artifacts of using MbCD.
5. I'm wondering if the authors should add a bit of discussion putting their work in the context of other "pre stimulation" priming work in T cells. For example, there is a notable body of literature on whether molecules such as TCR, LAT etc are preclustered at the synapse, mainly also stemming from super-resolution microscopy – some conclusions are still controversial.

Title: CD45 pre-exclusion from the tips of T lymphocyte microvilli prior to antigen recognition

Authors: Jung et al.

REVIEWER COMMENTS

Reviewer #1 (Remarks to the Author):

The authors use expansion microscopy and Airyscan microscopy to achieve 40 nm resolution of protein distributions in small projections on T cells that are widely referred to as microvilli. They show that L-selectin and TCR are biased to microvilli tips as previously described using TIRF imaging by some the first author. The authors here also show this by dSTORM. The authors convincingly show that under the experimental conditions CD45 is excluded from the microvilli tips and that normal levels of plasma membrane cholesterol are critical for the CD45 exclusion from the tips, but that the CD45 extracellular domain is not required. This result is different from the TIRF analysis from Haran's lab from 2016 and a more recent paper from Haran lab what argues that CD45 distribution seemed to be uniform. There are potential issues with the conditions that need to be considered (see below). The data on Calcium imaging and proof that the Calcium flux is induced by microvilli contacts is not convincing as presented, but with controls and further analysis this seems like a productive direction. The working model presented is quite confusing as they start with a one-sided presentation of kinetic segregation as a model to explain TCR triggering and then conclude that CD45 is already segregated from the TCR at microvilli tips, which if true would largely invalidate the simplest version of the K-S model.

Major issues:

1. Fixing the cells after holding them at 4 degrees is a problem for this experiment as work from Baird and colleagues has shown that cooling cells results in generation of microscopic liquid ordered domains that cannot be detected at 37 degrees. See Baumgart T, Hammond AT, Sengupta P, Hess ST, Holowka DA, Baird BA, Webb WW. Large-scale fluid/fluid phase separation of proteins and lipids in giant plasma membrane vesicles. *Proc Natl Acad Sci U S A.* 2007;104(9):3165-70. Epub 2007/03/16. doi: 10.1073/pnas.0611357104. PubMed PMID: 17360623. This has been explicitly demonstrated for T cells- Magee AI, Adler J, Parmryd I. Cold-induced coalescence of T-cell plasma membrane microdomains activates signalling pathways. *J Cell Sci.* 2005;118(Pt 14):3141-51. Epub 2005/07/15. doi: 10.1242/jcs.02442. PubMed PMID: 16014381. It would be much better if the labelling could be done with monovalent, non-triggering/inhibitory reagents at 37 degrees and then the cells fixed at 37 degrees prior to analysis. I suspect that the liquid ordered domains are formed at the tips of microvilli at low temperature, and this probably means something biological based on the Magee et al paper, but I expect that

CD45 will not be excluded from microvilli tips at 37 degrees. Certainly, Cai et al 2017 didn't see any evidence of this and neither of the studies from Haran lab, but perhaps the resolution was inadequate in those situations.

We thank the reviewer for this important comment. In our previous experiments, we labeled and fixed samples at 4 °C for human and mouse T cells and B cells, but Jurkat cells were labeled and fixed at room temperature (RT) for 15 minutes (result summarized in Fig. 6 and Supplementary Figs. 9 and 10). Under these conditions, endogenous CD45 was clearly depleted from MV tips. We have now repeated these experiments labeling the cells at 37 °C for 15 minutes (described in p. 8 of the revised manuscript and in Supplementary Fig. 2). We found clear exclusion of CD45 from the tips and highly enriched L-selectin at the MV tips. Statistical comparison of samples labeled at the 4 °C *versus* 37 °C did not show significant differences for SR' values and intensities in the segmented areas, except in the MV-column areas. The RIS, (Radiometric Intensity Score, previously defined as GP) differences between the segmented areas were slightly larger in the sample prepared at 37 °C. It appears that CD45 exclusion was more extended to the Tip-col areas when cells were labeled at 37 °C. However, the depletion in the CD45 MV-tip area was clearly not significantly altered. We conclude, therefore, that CD45 exclusion from the MV-tips was not the result of cooling at 4 °C. In addition, it has been reported that the high concentration of cholesterol or glycosphingolipids increases the stability of the lipid domains (Refs. # 48, 49, 53), consistent with our hypothesis that CD45 segregation is causally related to cholesterol enrichment in the lipid domains.

2. The ODT/Calcium measurement approach clearly has promise, but the results are preliminary. Many papers have shown that calcium flux precedes stable synapse formation since the mid 1990's and maybe even earlier, but this is quite explicitly stated and examples are shown in the Bunnell et al 2002 paper and in the Campi et al 2005 paper. But it has not been well documented that these are contacts with microvilli. It looks like this is going in the correct direction, but just needs better quantification and controls.

In response to your comment, we quantitatively analyzed the data in four groups of cells:

1. OTII-Rag2^{-/-} CD4⁺ T cells that are NOT in contact with a B cell
2. OTII-Rag2^{-/-} CD4⁺ T cells that ARE in contact with a B cell
3. OTII-Rag2^{-/-} CD4⁺ T cells that are NOT in contact initially, but acquire a B cell contact during the acquisition
4. OTII-Rag2^{-/-} CD4⁺ T cells incubated with B cells without antigen

We now show the data for groups 1, 2 and 3 (Supplementary Fig. 11), demonstrating highly significant differences. Group 3 is the most informative, because we can relate the calcium flux with the initiation of the contact. The reviewer is correct that the calcium flux PRECEDES the formation of a stable mature synapse in 7 of 10 cells, starting when microvilli come in contact (Supplementary Video 3, cell 1).

The results of this re-analysis are now summarized in Supplementary Fig. 11 and described in p. 13 of the revised manuscript. As a specificity control, we analyzed T cells mixed with B cells that were NOT pulsed with antigen (group 4), and found that the intensity-time-trajectory of the contacting OTII-Rag2^{-/-} CD4⁺ T cell was not

significantly different to that of the non-contracting cells (Supplementary Video 4). We now discuss the Bunnell et al., and Campi et al. papers (p. 13, Ref. # 63, 64).

Minor issues:

1. Statistical analysis of a parameter related to TCR clustering and CD45 exclusion from the tips would be important. The general polarization value is misnamed as it contains no information about spatial polarization, which would tip/shift comparison. They could say that they mean “polarization” of the pixel intensity of the parameter, but this seems to be asking for confusion. The parameter is really a “signed segregation” score that goes to 0 when both molecules have similar normalized intensity and is +1 when only X is present and -1 when only Y is present. The spatial information only comes into play when the parameter is plotted against distance from the tip. A general polarization score might be developed by taking a ratio of tip vs. column, but this would require a decision as to where to draw this line between these two compartments. The use of the Pearson’s correlation coefficient is also help as its not biased by any author decision. But I feel that a parameter that capture the tip/column ratio and enable statistical testing on this conclusion is important.

Following the reviewer’s suggestion, we have made significant changes in image analysis. First, we updated segmentation by sub-dividing the MV area in the original manuscript into MV-tip area and MV-col area. The MV-tip area defines the region, in which pixel distances were <150 nm from each MV tip. Statistical comparisons between the MV-tip area, MV-col area, and CB area are newly added.

The ‘GP’ term was taken from the original reference (Ref. #45). We have replaced this term with ratiometric intensity score (RIS). To calculate RIS, the intensity of second channel (ch2) image was normalized to match the mean intensities between ch1 and ch2. This satisfies the reviewer’s suggestion that RIS “goes to 0 when both molecules have similar normalized intensity and is +1 when only X is present and -1 when only Y is present”.

We found that the use Pearson’s correlation coefficient in the segmented area is inappropriate. For example, if all pixel intensities of chX in the segmented area are 0 (100% exclusion of molecule X) while all pixel intensities of chY in the segmented area are equal to maximum intensity (100% enriched molecule Y), the variance of $(X_i - \bar{X})(Y_i - \bar{Y})$ (Supplementary Table 1) become zero instead of -1 (as explained in p. 6). Instead, we analyzed SR', the modified version of Pearson’s correlation to show the correlations of ch1 and ch2 images. It is still intensity insensitive and unbiased on the author’s decision.

The CD45 column/tip intensity ratio analysis has been updated in Fig. 4j and Supplementary Fig. 2o.

2. The introduction is biased toward kinetic segregation model as a primary triggering mechanism. Some of the earliest studies on CD45 knockout T cells showed that function can be restored by the forms of CD45 lacking the extracellular domain (Volarevic S, Niklinska BB, Burns CM, June CH, Weissman AM, Ashwell JD. Regulation of TCR signaling by CD45 lacking transmembrane and extracellular domains. Science. 1993;260(5107):541-4. PubMed PMID: 8475386.). The authors discuss a general exclusion of CD45 from the entire cell interface, but

application of TIRF to T cell activated on substrates demonstrated that CD45 exclusion was very local to TCR clusters, which are likely collapsed microvilli (Varma R, Campi G, Yokosuka T, Saito T, Dustin ML. T cell receptor-proximal signals are sustained in peripheral microclusters and terminated in the central supramolecular activation cluster. *Immunity*. 2006;25(1):117-27. Epub 2006/07/25. doi: 10.1016/j.immuni.2006.04.010. PubMed PMID: 16860761). Finally, a recent paper using nanofabrication methods demonstrated that preventing CD45 exclusion results in a greater requirement for receptor clustering, but didn't prevent triggering (Cai H, Depoil D, Muller J, Sheetz MP, Dustin ML, Wind SJ. Spatial Control of Biological Ligands on Surfaces Applied to T Cell Activation. *Methods Mol Biol*. 2017;1584:307-31. doi: 10.1007/978-1-4939-6881-7_18. PubMed PMID: 28255709.). There is also the work from Weiss on the importance of CSK. So I think there are a number of ways to think about triggering. Certainly, if the authors continue to find CD45 exclusion even at 37 degrees then it would be ideal to be less biased, which make it easier to consider how the system could function with CD45 already segregated from the TCR.

Thanks for these comments. For the first point raised by the reviewer, we now describe in the introduction (p. 3) the role of CD45 in T cell activation. For the second point, we agree that the previously observed CD45 in the TIRF system could be due to collapsed MVs, however, spatial resolution is largely limited in xy in TIRF, thus we prefer to avoid including this assumption. We now cite in the introduction part (p.3) two recent studies using super-resolution microscopy, which revealed that CD45 exclusion is initiated from MV-mediated close contacts that expand as the IS is maturing (Ref. # 16, 17), consistent with the reviewer's comment. For the third point, we added to the discussion (p. 14) a relevant reference (Ref. #73). The method introduced in the nanofabrication paper can only prevent CD45 exclusion driven by extracellular domain. As we show here, the pre-existing CD45 exclusion was driven by the transmembrane domain, not by that extracellular domain. A similar study published by the same authors (*Nat Nanotechnol*. 2018; 13: 610–617) indicated the importance of the spatial positioning of ligand for triggering T cell signaling. Particularly on 3D surfaces, close packing of engaged TCR at inter-TCR spacing of < 50 nm was required for robust signaling and, in fact, in this system CD45 was weakly excluded. This makes an interesting point that TCRs are actually concentrated at the tip. Therefore, although we do not know the exact mechanism, CD45 exclusion and CD3 positioning at the tip could be major key players that control the TCR triggering threshold. For the 4th point, the relevant reference (Ref. #83) has been added.

Reviewer #2 (Remarks to the Author):

Here, the authors use expansion microscopy to analyse the distribution of CD45 in microvilli at the T cell immunological synapse. It has been widely postulated that CD45 needs to be excluded from sites of TCR engagement in order to facilitate signaling but this is the first work showing that the phosphatase is pre-excluded from the very initial contact sites. Overall, I think this is an important finding and I support publication. The paper is also technically quite sound and so I don't think it needs that much work to bring it to a publishable standard. Overall then I recommend publication subject to the following corrections, which I don't think are too difficult.

We thank the reviewer for the highly positive comments.

1. There is a lack of image quantification and statistical testing, even starting from the very First Figure. The authors should apply a method of colocalization (e.g. Pearson's) to the microvilli tips and sides and show that they are indeed statistically significantly different. That is the case for all claims of differential colocalization throughout the manuscript.

We appreciate this valid suggestion. As mentioned in response to Reviewer #1, we have updated image analysis and included statistical analysis of the segmented areas. The quantification results of L-selectin vs. CD45 (Fig. 1) are updated in Supplementary Figs. 2, 3, and 6. The explanation for the quantification approach, which was originally described in the context of Fig. 2, has been moved to pages 6-7 of the revised manuscript. Colocalization is analyzed using the SR' method (See the response to Reviewer #1).

2. To me, the STORM experiments don't add much understanding. If I were them, I'd leave this out, or, if they stay in they need to be quantified by some method of colocalization too.

For the STORM data colocalization analysis, we calculated the pair-correlation of the STORM data in Supplementary Fig 5 (as explained in p. 8). Note that intensity of the STORM data is not linear to the molecular density (stochastically multiple blinking happens from a single molecule), thus generally do not use the classical colocalization method for STORM images; instead pair-correlation analysis is more commonly used.

3. For Fig. 2, I don't really understand the motivation for the use of the GP measure – rather than a simple ratio or other method of colocalization such as Mander's? Could the authors explain why they chose this procedure and how statistical significance can be inferred from it?

The RIS term (GP in the original manuscript) is not only beneficial for demonstrating clear CD45 exclusion, but it also provides biologically valuable information, particularly for the CD3:CD45 ratio that is directly linked to TCR activation. The RIS analysis is particularly necessary to interpret SR' result. For instance, SR' can be -1 in both cases: $ch1=0$ and $ch2= \max$; $ch1=\max$ and $ch2=0$. Now the test for the statistical significances are updated for RIS and SR' for the colocalization analysis. Besides, simple ratio ($ch1/ch2$) can be problematic since the range of the result will be too varied and zero intensities cannot be a divisor. The RIS is the radiometric calculation that calculate relative ratio between the two channel -1 to 1.

4. The hypothesis that cholesterol accumulates at the tips seems out of the blue – why did the authors hypothesise this – is there literature on the subject? The use of MbCD has been problematic in some cell types – possibly creating gel phases etc. and some, such as the Gaus lab have used 7-ketocholesterol as an alternative for ablating the membrane ordered phase. I think the authors should consider this, but I'd be happy with some discussion on the potential limitations/artifacts of using MbCD.

There are two main reasons why we tested the effect of cholesterol depletion on CD45 exclusion. First, cholesterol is a major component of lipid rafts (Ref. #27-29) and its content is a critical determinant of membrane thickness (Refs. #27, 33-35). Second, cholesterol is known to induce curvature by asymmetric accumulation on inner leaflets to fill gaps in the acyl chains lipid bilayer (Refs. #30-32). Although the use of M β CD to treat cells can be problematic, it has been widely used in studies of primary human T cells and Jurkat T cells (Refs. #56-58) showing clear depletion of cholesterol and disruption of lipid rafts. Treatment with 10 mM of M β CD also resulted in increased membrane fluidity in primary human T cells (Ref. #56). Although biased sensitivity M β CD treatment (1 mM) to cholesterol in the liquid-disordered phase has been reported (J. Membr. Biol.; 2011 241:1-10), this selectivity is diminished at a higher M β CD concentration. It has also been reported that M β CD similarly extracts cholesterol from lipid raft or non-lipid raft domains in Jurkat cells (Ref. #58). Therefore, we believe that treatment of 10 mM M β CD to extract cholesterol is appropriate. We feel the reviewer's comment to the effect that M β CD can create gel-like phase (more ordered than liquid-fluid phase) is inconsistent with our results because, if that was the case, we would have observed increased, rather than decreased, CD45 exclusion.

5. I'm wondering if the authors should add a bit of discussion putting their work in the context of other "pre stimulation" priming work in T cells. For example, there is a notable body of literature on whether molecules such as TCR, LAT etc. are preclustered at the synapse, mainly also stemming from super-resolution microscopy – some conclusions are still controversial.

We are not sure whether we interpreted correctly the reviewer's comment regarding "pre stimulation priming work in T cells". In describing "pre-stimulation" priming work in T cells, we referred to the pre-existing clusters observed before antigen stimulation and IS formation. As requested, we have added additional discussion in the revised manuscript (p. 15, Ref. # 23, 74-76). Pre-clustering of molecules such as TCR or LAT on the cell surface (but not at the synapse *per se*) before priming has been supported by several super-resolution studies, including ours. We do not see how pre-clustering can occur at the synapse since a T cell would not "know" where a synapse will be formed prior to its contact with an APC. To us, pre-clustering means formation of a cluster before stimulation, thus, formation of these pre-clusters is driven by physical properties of a given molecule or the membrane, rather than by extrinsic stimulation.

We believe that our rebuttal properly and completely addresses the reviewers' comments, and we hope that you will now find our revised manuscript acceptable for publication. Thank you very much.

REVIEWERS' COMMENTS

Reviewer #1 (Remarks to the Author):

The authors have largely addressed my concerns and presented a balanced discussion. The authors have done the 37 degree experiments with CD45 and Lselectin. I would have liked to see this for CD45 and TCR, but I give that it seems like the CD45 and Lsel compartments pretty much fill cover the microvilli and the TCR co-localizes with Lsel at low temperature the idea that TCR is segregated from CD45 as physiological temperature is strongly suggested. They have also now cited the Magee et al paper and other counter-examples, so readers will be aware of the limitation. So I have no further concerns. I congratulate the authors on a visually impressive study that I'm sure will stimulate the field.

Reviewer #2 (Remarks to the Author):

The authors have addressed my concerns. From my side I think the paper is acceptable for publication, but I note Reviewer1 had more technical concerns.